

# A data-driven framework for assessing climatic impact-drivers in the context of food security

Marcos Roberto Benso[1], Roberto Fray Silva[2], Gabriela Gesualdo Chiquito[1], Antonio Mauro Saraiva[2], Alexandre Cláudio Botazzo Delbem[4], Patricia Angélica Alves Marques[3], and Eduardo Mario Mendiondo[1]

[1]São Carlos School of Engineering, University of Sao Paulo, Sao Carlos-SP, 13566-590, Brazil
[2]Institute of Advanced Studies, University of São Paulo, São Paulo, SP, Brazil
[3]Luiz de Queiroz College of Agriculture, University of São Paulo, Piracicaba, SP, Brazil
[4] Institute of Mathematics and Computer Sciences, University of Sao Paulo, Sao Carlos-SP, 13566-590, Brazil

**Correspondence:** Marcos Roberto Benso (marcosbenso@gmail.com)

**Abstract.**

Understanding how physical climate-related hazards affect food production requires transforming climate data into relevant information for regional risk assessment. Data-driven methods can bridge this gap; however, more development must be done to create interpretable models, emphasizing regions lacking data availability. The main objective of this article was to evaluate

the impact of climate risks on food security. We adopted the climatic impact-driver (CID) approach proposed by Working Group I (WGI) in the Sixth Assessment Report of the Intergovernmental Panel on Climate Change (IPCC). In this work, we used the CID framework to select the most relevant indices that drive crop yield losses and identify important thresholds for the indices. When these thresholds are exceeded, the impact probability increases. We then examine the impact of two CID types (heat and cold, and wet and dry) represented by indices of climate extremes considering the impact on different crop

yield datasets, focusing on maize and soybeans in the central agro-producing municipalities in Brazil. We used the random forest model in a bootstrapping experiment to select the most relevant climate indices. Then, we applied the Shapley Additive Explanations (SHAP) with the XGBoost model explanatory analysis to identify the indices thresholds that caused impacts. We found that the mean precipitation is a highly relevant CID. However, there is a window in which crops are more vulnerable to precipitation deficit. For soybeans, in many regions of Brazil, precipitation below 80 mm/month in December, January, and

February represents an increasing risk of crop yield losses. This is the end of the growing season for those regions. In the case of maize, there is a similar pattern with precipitation below 100 mm/month in April and May. Indices of extremes are relevant to represent crop yield variability. Nevertheless, including climate means remains highly relevant and recommended for studying the impact of climate risk on agriculture. Our findings contribute to a growing body of knowledge critical for informed decision-making, policy development, and adaptive strategies in response to climate change and its impact on agriculture.





# 1 Introduction

Climate extremes such as heat waves, droughts, floods, and excessive precipitation are crucial in determining crop yield short-falls (Vogel et al., 2019). Based on empirical evidence from several studies, models that use data from multiple weather variables are more accurate in describing the variability of crop production than models that only use precipitation (Proctor
et al., 2022; Ray et al., 2015) or single weather variables. Consequently, it is critical to assess the risk of agricultural production through extreme climate indices, which are a fundamental part of monitoring hazards that can cause impacts on food production and food security (Das et al., 2022; Schyns et al., 2015).

Several natural hazards impact society; therefore, they can be named "impact-drivers". According to Ruane et al. (2022), it is only possible to understand the impact-drivers by knowing the vulnerability and exposure of the specific sector. Sectoral
information can determine the magnitude of the driver's effect, which can be beneficial or detrimental to its activities. This requires a co-creation process that aims at contextualizing climate information for decision-making. This concept was implemented as the climatic impact-drivers (CID) framework. It was introduced by Working Group 1 of the Intergovernmental Panel on Climate Change (IPCC) in the Sixth Assessment Report (AR6) (Ranasinghe et al., 2021; Ruane et al., 2022).

The formal definition of the CID is the physical conditions of the climate that include means, extremes, and events. The
CID is framed in terms of two critical aspects of risk assessment, defining *"Indices for climatic impact-drivers"* and identifying *"Thresholds for climatic impact-drivers"* (Ruane et al., 2022). These aspects reinforce the learning necessary for determining numerically computable indices that utilize one or a combination of climate variables to quantify the intensity and frequency of a CID. When surpassed, the values of the indices increase the danger of losses and damage.

Although many indices have been used in the literature and summarized by Ranasinghe et al. (2021) regarding their rele-
vance to the agricultural sector, it is still necessary to tailor the approach based on regional characteristics that bridge global information with local solutions. The CID framework is still in its early stages of development, as it is in line with the United Nations Sendai Framework for Disaster Risk Reduction (UNISDR) 2015–2030 (UNDRR, n.d.) and is following the definitions of the UNISDR hazard list. However, the CID framework also considers climate change a significant hazard, which is not included in the UNSIDR hazard list.

However, the lack of data and the interpretability of the models are two challenges shown in the literature. One of the main challenges in evaluating the impact of weather extremes on food production and security is the limited availability of high-quality yield and weather data at suitable temporal and spatial scales (Vogel et al., 2019).

Reanalysis datasets offer extended time series with high spatial and temporal resolutions. However, reanalysis models can produce high uncertainty when estimating poorly monitored regions (Condom et al., 2020). The European Centre for Medium-
Range Weather Forecasts ERA5 weather dataset has been recommended for agricultural applications Almendra-Martín et al. (2021). Nevertheless, more research is needed to investigate the applicability of ERA5 in crop yield responses to climate variability studies.

The paradigm of interpretability of machine learning models is a broad topic of discussion in supervised learning (Lipton, 2018). Two essential observations related to model interpretability are: (i) to better interpret the causal association between



data, correlation methods could be used; and (ii) the training data can be imperfect to represent a dynamic environment that changes over time.

One way to understand climate change's impact on agricultural production is through Machine Learning (ML) Algorithms. ML algorithms can improve our understanding of the impact of climate on crop yields (Sidhu et al., 2023). Drawing on statistical learning theory (Vapnik, 1999), these algorithms can generalize patterns and make predictions from available data.

Several authors have applied ML algorithms to predict crop yields (Van Klompenburg et al., 2020; Vogel et al., 2019; Sidhu et al., 2023; Han et al., 2019; Pantazi et al., 2016; Schierhorn et al., 2021; Silva Fuzzo et al., 2020). In the study of Silva Fuzzo et al. (2020) a prediction model was employed to evaluate the variability of soybean crop yields in Paraná. They used satellite-derived surface radiant temperature (Ts) and the Normalized Difference Vegetation Index (NDVI), coupled with water balance performed using satellite evapotranspiration and precipitation data, along with reanalysis. The study findings demonstrated a

strong alignment between the model predictions and actual crop yields. ML was also essential in the works by Pantazi et al. (2016) and Han et al. (2019).

Decision tree algorithms such as random forest (RF) models have been used to improve understanding of the impact of weather extremes on crop yield variability (Vogel et al., 2019; Jeong et al., 2016; Schierhorn et al., 2021). The RF model combines tree predictors that are split recursively and used for predictions (Breiman, 2001). It can provide information on

each feature's importance in the model's overall performance. This was used by (Vogel et al., 2019) and (Schierhorn et al., 2021) to unravel the influence of extreme temperature and precipitation from mean climate variables on predictive RF models of soybean and maize. The two articles agree that mean climate variables over growing seasons are the most relevant features for predicting crop yields. However, extreme weather indices, especially for droughts and temperature extremes, can explain 18-43% of crop yield variability (Vogel et al., 2019).

Despite the effort to increase the performance and interpretability of ML models, the studies cited previously relied on a somewhat limited selection of indices. This means other factors influencing crop yield variability can remain hidden, and the underlying mechanisms of crop yield losses due to weather extremes can be neglected.

A generalized framework for variable and feature selection has been designed to improve the performance of ML models, provide faster models, and improve understanding of the underlying processes that generated the data (Guyon and Elisseeff,

2003). The backwards recursive feature elimination (RFE) presented by (Svetnik et al., 2004) uses the ability of RF algorithms to generate variable importance as a variable reduction wrapper algorithm. Several articles have shown its applicability for suitable crop selection (Wang and Li, 2023) and hyperspectral imaging to monitor pasture quality (Pullanagari et al., 2018).

Although eliminating redundant features and variables can improve our understanding of the data structure, the problem of interpretability cannot be fully addressed. We propose using the model-agnostic explanation method introduced by (Lundberg

and Lee, 2017) called SHAP (SHapley Additive exPlanations). There are some promising studies applying SHAP to environmental data (Wikle et al., 2023; Viana et al., 2021) with implications for soil moisture and determination of evapotranspiration. The use of *post hoc* explanation algorithms for crop yields was used by Mariadass et al. (2022). Nevertheless, to our knowledge, this method has not been specifically applied to predict the impacts of extreme weather on food production.




We introduce a comprehensive modeling framework to enhance the interpretability of tree-based models that utilize climate
data to predict crop yield losses. This framework incorporates filtering methods for feature selection and determination of
variable importance, along with *post hoc* analysis aimed at inferring the impact of each selected variable on the prediction.

In this research, our main objective is to assess the impact of climate extremes on food production. To achieve this, we have
used the climatic impact-driver (CID) framework developed by Working Group 1 (IPCC WG1) of the Intergovernmental Panel
on Climate Change (IPCC). This framework allows us to characterize climate extremes by creating numerically computable
indices and determining relevant thresholds. The significance of this framework lies in its ability to provide a basis for incor-
porating climate information into studies, decision-making processes, and policy development. By applying this framework
to our research, we aim to provide valuable insights that can inform critical decisions, policies, and strategies related to food
production in the face of climate extremes.

## 2 Methodology

In this section, we present a framework for investigating the impacts of climate extremes on crop yields. The framework aims to
investigate the impacts of climate extremes on crop yields using ML, focusing on building a reproducible workflow, selecting
features, and producing explainable model outputs. A good feature of an ML algorithm is **relevant** to explain the target variable
(in this study, crop yields) based on the input variables. However, it should not be **redundant** with any other relevant predictor
(Yu and Liu, 2003). In addition to these concepts widely applied in ML methods, we add the concepts of **explainable** and
**operational** features. In ML, a feature is any variable used as an input variable for prediction. Therefore, this work will use the
terms feature and variable interchangeably.

### 2.1 Modeling Framework

This section introduces a modeling framework that seeks to understand the main climatic influences on soybean and maize crop
yield and determine the thresholds of these climatic influences using an entirely data-driven approach (Fig. 1). This framework
consists of three steps. The first is data filtering; in this step, we removed highly correlated features (Pearson correlation
greater than 0.9). Feature selection is considered to be a pre-processing step in machine learning models. The filtering process
removes redundant features. Other relevant aspects, such as relevancy, explainability, and operationability, will be explained in
the following steps. The second step aims at selecting the most important variables. We use the abilities of the random forest to
generate the importance of variables to rank the most important variables. The third step is to define variable thresholds. To do
that, we must apply another machine model to explain the first one. The Shapley Additive Explanations (SHAP) explanatory
analysis is an explanation algorithm proposed by Štrumbelj and Kononenko (2014) and uses game theory (Strumbelj and
Kononenko, 2010) to provide an efficient explanation of the predictions made by a machine learning algorithm. The SHAP
method uses a second model, most commonly the eXtreme Gradient Boosting (XGBoost) model (Ribeiro et al., 2016), to
explain how each variable was used to make each model prediction.





**Figure 1.** Flowchart illustrating the methodology proposed for analyzing the impact of climate indices on crop yield.

In the second step, we identify the most critical climatic impact-drivers (CID) to assess their impact on food production. To achieve this, we used the random forest model. Different models were trained considering different combinations of input data, including precipitation means, temperature means, and combinations of means and extreme climate indices. The goal of this experiment was to identify the most important climate indices.

    Random Forest (RF) algorithms are a combination of multiple classification and regression trees (CART) predictors that are

sampled independently and with the same distribution (Breiman, 2001). This model has a wide range of applications and has





gained much attention for predicting the impacts of climate on regional crop yields (Jeong et al., 2016). Regarding the first step, redundancy is not a problem for tree-based algorithms because algorithms are not built under the assumption of a fixed functional relationship between the predictor variables (Chan et al., 2022). For the sake of model performance, it might not be the problem; however, it can be a problem for the interpretation of machine learning models (Aydin and Iban, 2023).

RF algorithms are based on a model that creates many trees through bootstrapping. The algorithm creates bootstraps from the original training datasets and then resamples different subsets of randomly selected features to make predictions using a wide range of decision trees or splits. Each prediction is then tested with a remaining part of the data not selected for the bootstrap dataset, also called the Out-of-Bag (OOB) dataset. More accurate decision trees receive a higher weight in the model and are aggregated. However, split-variable randomization is implemented to avoid highly correlated trees, limiting the number

of split variables to a value. For training and testing random forest models, we used the R Package ranger (Wright and Ziegler, 2017). The performance of the models was measured using the coefficient of determination. To summarize the results, we calculated the performance for each state considering the median and the upper and lower 95 % confidence intervals. Medians and confidence intervals were calculated using R-package QuantileNPCI (Hutson et al., 2019), which implements Hutson's algorithm to calculate nonparametric confidence intervals for quantiles (Hutson, 1999).

The third step builds on the results of the second step and uses the most relevant climate indices employing an XGBoost explainability with Shapley Additive Explanations (SHAP) explanatory analysis. This approach aimed to provide a detailed understanding of how the model used these crucial indices and attempted to identify significant thresholds for these influential climate variables. The XBoost models were implemented using the R package xgboost (Chen et al., 2023), and the SHAP explanations were implemented using the R package shapviz (Mayer, 2023).

The eXtreme Gradient Boosting, also called XGBoost, was created by Chen and Guestrin (2016) and is a variant of the boosting tree algorithm family. Boosting tree and random forest algorithms have the same principle of combining the outputs of individual decision trees; however, the way each tree is built is different between the two, and this also affects the way the outputs of each tree are combined (Sutton, 2005). Boosting algorithms build trees sequentially rather than randomly; this means that the algorithm trains a single tree and then evaluates the quality of this tree using a penalty function, so each new

tree tries to correct the mistakes of the previous one in the form of optimization problem (Breiman, 1997). This optimization problem gives a weight to each created tree, and the prediction is the sum of all trees (Sutton, 2005). XGBoost is considered to provide state-of-the-art results for agricultural applications (Mariadass et al., 2022), and its complexity and set of equations can be further explored in Chen and Guestrin (2016).

The SHAP approach is based on explaining how each feature of the model was used to make a single prediction. The first

step is to define a base prediction, which in this case is the expected value of the set $X$, which is the values of the adjusted and detrended crop yields $E[f(X)]$. Then, the XGBoost algorithm is used to make a prediction $f(x)$ for a single value of crop yield in a specific municipality and a particular year. The difference between the expected value and the prediction is called the SHAP value and preserves the unit of crop yields; therefore, here, it will be in tons per hectare.

In order to identify how each feature was used to generate the SHAP value, the algorithm recursively adds each feature

and tests the importance of the feature for that prediction. However, the order in which the feature is added to the model is





essential; that is why the principles of game theory introduced by Lloyd Stowell Shapley were used to solve the problem of allocating each feature's order and extracting its importance for the prediction. More details of the game theory used in SHAP explanations can be learned in Strumbelj and Kononenko (2010).

In Fig. 2, we provide a graphical illustration of the black box approach in which the XGBoost algorithm uses three fea-
tures, Temperature (Temp), Precipitation (Prec), and Standardised Precipitation Evapotranspiration Index (SPEI) to make a prediction. The difference between base prediction and the model output, also called SHAP Value, is explained with SHAP explanations.

As we perform the SHAP explanations for each prediction, we can generate a partial dependence plot, which is the relation-
ship between the feature value and the contribution to the SHAP value. From this approach, we can unveil which feature values
are critical for crop yield losses, establish thresholds, and contribute to the CID framework. Since the order of each feature is also evaluated, a second analysis is performed, which uses Gaussian copulas to evaluate the combined effect of the variables.

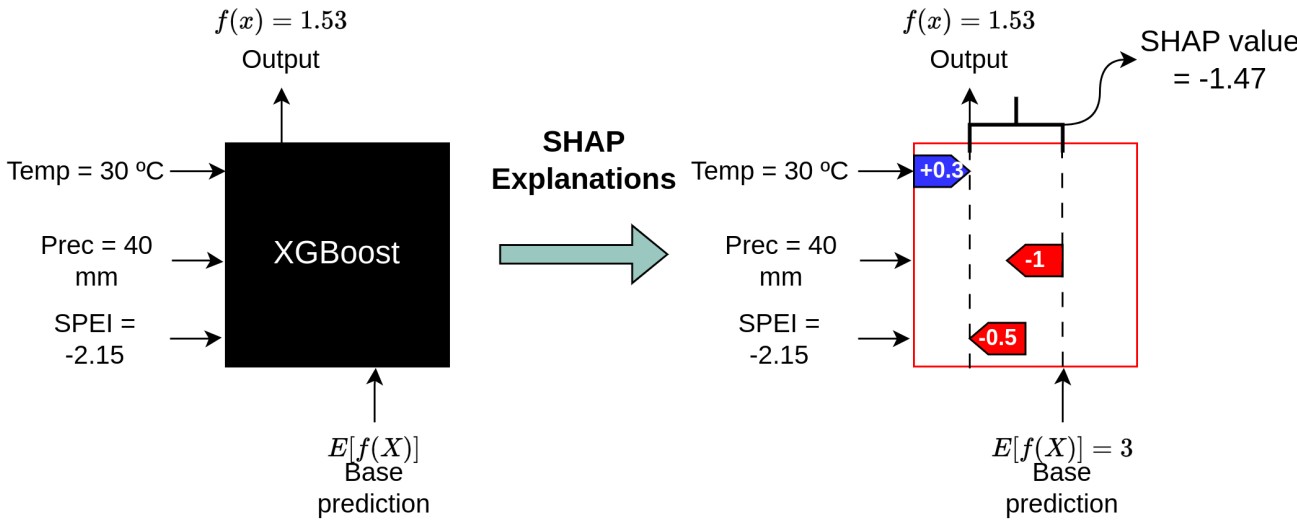

**Figure 2.** Demonstration of SHAP Explanations. Adapted from Lundberg and et al. (2023)

## 2.2   Study Area

The richness of plant biodiversity offers a wide range of possibilities for human nutrition. Currently, the existence of around 374,000 known and described plant species is accepted (Christenhusz and Byng, 2016). According to Rapoport and Drausal
(2013), more than 16,000 species are considered edible. The diets of most of the human population are based on one or more of the main staple crops, such as soybeans (Hartman et al., 2011) and maize (Shiferaw et al., 2011).

Although we recognize that food production and food security require studying a broader range of plants and animal prod-
ucts, in this work, we will examine the second season of the production of soybean and maize. These two products represent the main staple food products in terms of area and quantity.



Brazil is a significant producer of agricultural goods, as reported by the Food and Agriculture Organization (FAO) (FAO, n.d.). This country is responsible for more than 10% of the world's maize and more than 30% of the global soybean production. Brazil is one of the four leading agricultural producers in the world, along with China, India, and the United States, with a cultivated area of soybean and maize of 58 million hectares. In Fig. 3, we show the delimitation of the study area. The map shows 452 selected municipalities that encompass the states of Rio Grande do Sul (RS), Santa Catarina (SC), Paraná (PR), São

Paulo (SP), Mato Grosso do Sul (MS), Minas Gerais (MG). The selection criteria will be explained in the sub-section Crop yield data. The map shows the percentage of cropland derived from the work of Potapov et al. (2022).

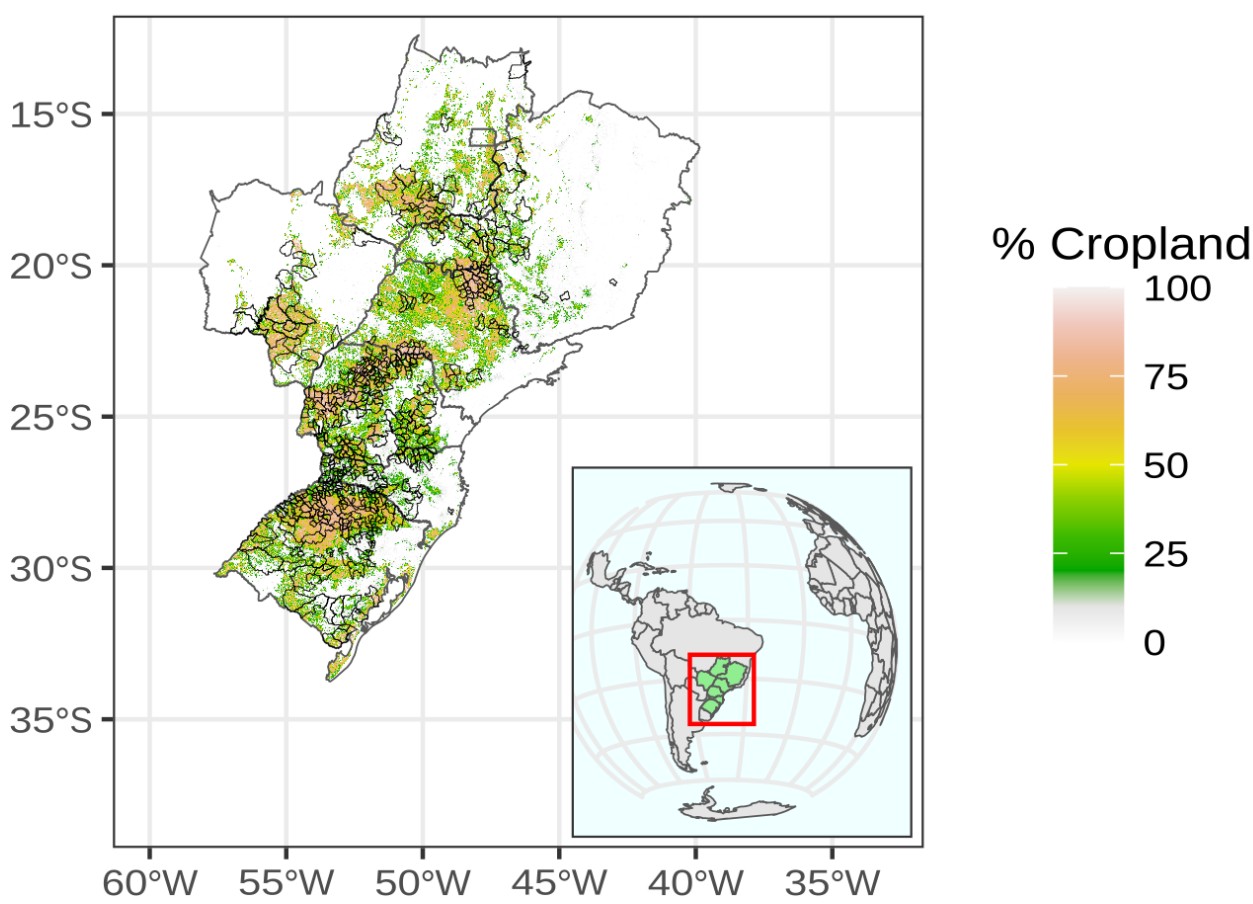

**Figure 3.** Location of selected study municipalities with respect with observed cropland extent between from 2003 to 2019

The growing season in the study area was defined using a global crop calendar for the second season of soybeans and maize determined by data from Sacks et al. (2010). We consider that the planting dates follow a normal distribution, with the mean date being the most probable date for farmers and the maximum and minimum dates being considered twice the standard



deviation. For soybeans, sowing dates start in the middle of Austral spring in October, peak in November, and end in December. The harvest begins in the late summer and extends to fall, from February to March. Since the second season of Maize peaks in February, we consider that the end of Soybean is in February. For the second season of maize, planting begins at the end of January, after soybean harvest, peaks in February, and ends at the beginning of April. The harvest starts in June, peaks in August, and ends in October.

## 2.3 Data collection and processing

In this section, we present the description of datasets used to analyze the impact of climate variables on Soybean and Maize crops in the state of Parana. We used two criteria to select a dataset: (i) the data must comply with FAIR principles (i.e., data must have findability, accessibility, interoperability, and reusability); and (ii) climate data must be updated frequently (ideally, with minimum daily update frequency).

We used three different datasets: (i) the statistical yearbooks of the state of Paraná (Parana, 2021); (ii) the Municipal Agricultural Production Survey by the Brazilian Institute of Geography and Statistics (IBGE) (de Geografia e Estatística, 2022); and (iii) Global Dataset of Historical Yields (GDHY) (Iizumi and Sakai, 2020). For climate analysis, we used data from the fifth generation European Centre for Medium-Range Weather Forecasts (ECMWF) ERA5 Land Reanalysis dataset (Muñoz-Sabater et al., 2021). We summarized the main characteristics of each dataset used in Figure 1.

| Dataset | Variable | Spatial resolution | Temporal resolution | Time frame |
| --- | --- | --- | --- | --- |
| Crop yields IBGE | Soybean | Municipality level | Annual | 1974-2022 |
| Crop yields IBGE | Maize | Municipality level | Crop season | 2003-2022 |
| Crop yields Paraná | Soybean and Maize | Municipality level | Crop season | 1997-2021 |
| Crop yields GDHY | Soybean and Maize | 0.5 degree cell | Crop season | 1981-2016 |
| ERA5 Land | Precipiation, temperature | 0.1 degree cell | Daily | 1950-Current |
| SoilGrids 2.0 | Clay, Silt and Sand | 250 m cell | - | - |

**Table 1.** Description of the datasets used in the case study

## 2.3.1 Crop yield data

Crop yield data are a vital component in understanding the impacts of climate on food production. Crop yields are generally made available at the municipality level. We used three data sets to analyze crop yields in Brazil. We collected crop yield data from the Brazilian Institute of Geography and Statistics (IBGE) at the municipal level (de Geografia e Estatística, 2022). In the study area, the double cropping system is widely adopted. It, therefore, represents a potential bottleneck because the IBGE
data from 1974 to 2022 is an annual aggregation of the total production of that crop within the municipality.



However, IBGE has started to collect maize in the first and second season since 2003. This matches the period during which the second maize season intensifies in Brazil. Data at the municipality level were filtered based on data availability. The missing years were removed from the dataset, and the municipalities with more than two years of missing data were disregarded. The selection resulted in 452 municipalities for soybeans that comprise the states of Rio Grande do Sul (RS), Santa Catarina (SC),
Paraná (PR), São Paulo (SP), Mato Grosso do Sul (MS), Minas Gerais (MG), and Goiás (GO) and 216 municipalities for maize second season for Paraná (PR), São Paulo (SP), Mato Grosso do Sul (MS), Minas Gerais (MG) and Goiás (GO).

The Department of Rural Economy (Deral) of the Paraná state, Brazil, is also responsible for collecting crop data at the municipal level. The method of collecting and processing data is similar to what is done by IBGE; therefore, a high level of redundancy is expected from these two datasets. This redundancy is necessary to validate data and remove outliers that might
reduce the quality of a model. The same number of municipalities selected using IBGE data was used in data from Deral.

The Global Dataset of Historical Yields is a global annual time series of 0.5 ° grid-cell estimates for maize, rice, wheat, and soybeans from 1981 to 2016. For each grid cell, crop yields are estimated in ton/ha based on Food and Agriculture Organization (FAO) country-level yield statistics and then corrected using the remote-sensed leaf area index (LAI), the fraction of photosynthetically active radiation (FPAR) and crop-specific radiation use efficiency derived from reanalysis. Crop areas
and crop calendars were derived from Monfreda et al. (2008) and Sacks et al. (2010). More details on the dataset are described in Iizumi and Sakai (2020) and Iizumi et al. (2014). The dataset was aggregated to the municipal level using zonal statistics in the terra package (Hijmans, 2023) in R Studio.

For statistical analysis, we removed the outliers of all crop yield datasets considering, for each year, neighboring munici-palities using the interquartile range (IQR). Changes in technology in seed production, fertilizers, and land management, also
known as technological trends (Liu and Ker, 2020) were removed by Local Polynomial Regression Fitting (LOESS) (Cleve-land et al., 2017). The residuals of the LOESS model were tested for heteroskedasticity. If heteroskedasticity was proved, it was removed using the method proposed by the methods proposed by Ozaki et al. (2008). For further information on the preprocessing of crop yield data, please consult the Supplementary Material.

### 2.3.2 ERA5 Land reanalysis dataset

Weather data was sourced from the fifth generation European Centre for Medium-Range Weather Forecasts (ECMWF) ERA5 Land Reanalysis dataset (Muñoz-Sabater et al., 2021). The data set has a 0.1 x 0.1 lon lat-lon grid and was aggregated to the municipal area to match the spatial discretization of SBY. The data collection spanned 1980 to 2023, with daily observa-tions. Weather variables included precipitation, maximum temperature, and minimum temperature. We used different climate indices to evaluate multi-hazard risks. Since mean climate conditions of precipitation and temperature are the most relevant
Moriondo et al. (2011), we considered monthly precipitation, maximum and minimum temperatures, total precipitation, and mean temperature over growing seasons.



### 2.3.3 Indices for Climatic Impact-Drivers

The WGI of the IPCC has presented the climatic impact-drivers (CID) as a new approach to assessing climate data to analyze their effects on society. CIDs are represented by numerically computable indices and categorized into several types. In this paper, we considered wet and dry and hot and cold CIDs. To calculate the indices, we first considered the indices indicated by the Expert Team on Climate Change Detection and Indices (ETCCDI), which is supported by the World Meteorological Organization (WMO) Commission for Climatology, the Joint Commission for Oceanography and Marine Meteorology (JCOMM), and the Research Program on Climate Variability and Predictability (CLIVAR). A summary of the indices used according to the type of CID is shown in Table 2.

We also considered two drought-related indices, the Standardized Precipitation Index (SPI) (McKee et al., 1993; McKee, 1995) and the Standardized Precipitation and Evapotranspiration Index (SPEI) (Vicente-Serrano et al., 2010). The SPI is based on the probability of monthly precipitation on different time scales, and it is recommended to be calculated with a time series of at least 30 years. The monthly time series must be fitted to a cumulative distribution function (CDF). We adopted the gamma distribution.

Then, the data is transformed to the standard normal distribution to calculate the SPI, a standardized value subtracting the transformed precipitation from the mean value and dividing by the standard deviation. The SPI can be calculated using different time scales representing previous meteorological conditions, typically 1 to 48 months. For agricultural applications, 3-month SPI is the most suitable Kim et al. (2019); however, Lam et al. (2022) has shown a counter-example in Kenya showing that longer time scales could also be applied for agriculture.

The SPEI is a more recent index that incorporates temperature in the calculation of SPI. A new step was added to the procedure, calculating the monthly potential evapotranspiration (PET) and the SPI using the same procedure described previously with the value of monthly precipitation minus monthly PET. PET was calculated using the Hargreaves method, which is calculated using maximum and minimum temperature and extraterrestrial radiation (RA) (Droogers and Allen, 2002).

The primary motivation for using distinct indices derived from the same fundamental data is to identify which features of the extremes are the most significant. Is it the magnitude of an extreme, the length of time it lasts, or values that are either above or below a certain threshold? Research such as Vogel et al. (2019) has demonstrated the importance of extreme events in understanding the variability of crop yields. We summarize all the indices according to CID type and category in Table 2.





**Table 2.** Description of the Climatic Impact-Drivers (CID) considered in this study and their respective Indices

| CID Type | CID Category | CID Index Abbreviation | CID Index Description |
|---|---|---|---|
| Heat and Cold | Mean air temperature | Temp | Monthly temperature mean |
| | | DTR | Daily temperature range: Monthly mean difference between maximum and minimum daily temperature |
| | Extreme heat | TX90p | Monthly percentage of days when maximum daily temperature is higher than the 90th percentile |
| | | TN90p | Monthly percentage of days when minimum daily temperature is higher than the 90th percentile |
| | | SU | Number of summer days: monthly number of days when maximum daily temperature is higher than 25 °C |
| | | TR | Number of tropical nights: monthly number of days when minimum daily temperature is higher than 20 °C |
| | | TXX | Monthly maximum value of daily maximum temperature |
| | | TXN | Monthly maximum value of daily minimum temperature |
| | Cold spell | TX10p | Monthly percentage of days when maximum daily temperature is lower than the 10th percentile |
| | | TN10p | Monthly percentage of days when minimum daily temperature is lower than the 10th percentile |
| | | TNN | Monthly minimum value of daily minimum temperature |
| | | TXN | Monthly minimum value of daily maximum temperature |
| Wet and dry | Mean precipitation | Prctot | Monthly precipitation sum |
| | Heavy precipitation | R10mm, R20mm | Monthly count of days when daily precipitation is higher than 10 and 20 mm. |
| | | Rx1day, Rx1day | Monthly maximum 1-day and 5-day precipitation |
| | Agricultural and ecological drought | SPEI 3, 6 | Standardised Precipitation and Evapotranspiration Index for 3 and 6 moths accumulations |
| | | SPI 3, 6 | Standardised Precipitation Index for 3 and 6 moths accumulations |

### 2.3.4 Soil data

The soil property data used in this study were sourced from SoilGrids 2.0, a globally recognized resource that offers standard-
ized soil profile data Poggio et al. (2021). SoilGrids offers soil data, including clay, silt, and sand, at a resolution of 250 meters,





which was employed for our analysis at a depth of 30 centimeters. The content of each soil component was combined at the municipal level to correspond with the crop yield data.

## 3  Results and Discussion

The comparison of the datasets used in this study is important to evaluate the reliability of the data. High-quality crop yield
data improves the calibration of crop growth models (Rosenzweig et al., 2014). However, they have a broader application in geosciences. Crop yield data is used to parameterize hydrological models in watersheds, especially in agricultural catchments, and improve soil moisture simulation (Sinnathamby et al., 2017). For water resource management, using higher quality crop yield data has improved global knowledge on the water-food-energy nexus (Ai and Hanasaki, 2023; Wang et al., 2023).

We compared crop yields at the municipal level in Brazil. We observed that the IBGE and Parná Deral data for soybeans and
maize are highly correlated; however, outliers were detected in both datasets. The outlier removal process improved the agreement between the two datasets, suggesting that eliminating data improved the dataset's quality. Since Deral is only available in Paraná, for the other states of Brazil, GDHY and IBGE were compared. The global dataset of historical yields aggregated at the municipal level has a weak association with the other datasets. This result confirms what was reported by Iizumi et al. (2014). The GDHY is based on satellite data collected from a fixed cropland map. In many regions of Brazil, there is a noticeable
increase in croplands, which can influence the estimation of GHDY. In addition, the exact location of the planted area within each municipality can vary from year to year.

| (a) | Maize IBGE | Maize Deral | Maize GDHY |
|---|---|---|---|
| Maize IBGE | 1.000 | 0.910*** | 0.474*** |
| Maize Deral | 0.910*** | 1.000 | 0.596*** |
| Maize GDHY | 0.474*** | 0.596*** | 1.000 |
| (b) | Soy IBGE | Soy Deral | Soy GDHY |
| Soy IBGE | 1.000 | 0.968*** | 0.434*** |
| Soy Deral | 0.968*** | 1.000 | 0.403*** |
| Soy GDHY | 0.434*** | 0.403*** | 1.000 |

**Table 3.** Correlation coefficients between three different crop yield datasets:(a) IBGE (n = 3845), Deral (n = 3120), and GDHY (15411) for maize; (b) IBGE (n = 20629), Deral (n = 3432), and GDHY (15406) for soybeans. Values represent the strength and direction of the relationships, with *** indicating statistically significant correlations with p < 0.001

## 3.1  Identifying Key Climate Impact-Drivers

We tested a variety of indices that measure mean precipitation, mean temperature, and extremes. We initially tested the machine learning technique by using various inputs to demonstrate its ability to illustrate crop yield variability in the states examined. In
Fig. 4, we demonstrate the model performance of the random forest model considering different input variables and datasets.



Taking the coefficient of determination, the climate variables explained the variability of soybean crop yields, on average, from 30 to 40% of IBGE, 25 to 45% for GDHY, and 30 to 35% for Deral. The climate was explained for maize from 30 to 40% for IBGE and Deral and from 20 to 45% for GDHY.

The coefficient of determination quantified the proportion of the variance in the crop yield data (dependent or target variable) that the random forest model can explain. The results are consistent with the values found in other similar studies, Ray et al. (2015) used municipal level data to quantify the impact of climate variability on yields using regression models and determined that in Brazil, climate variability explains 26–34% of soybean yields and 41% of maize yields. Vogel et al. (2019) used the same dataset as Ray et al. (2015) considering South America and applied a random forest model defining the values of 28 % for soybeans and 25% for maize. It is important to note that none of these studies separated maize in the first and second season.



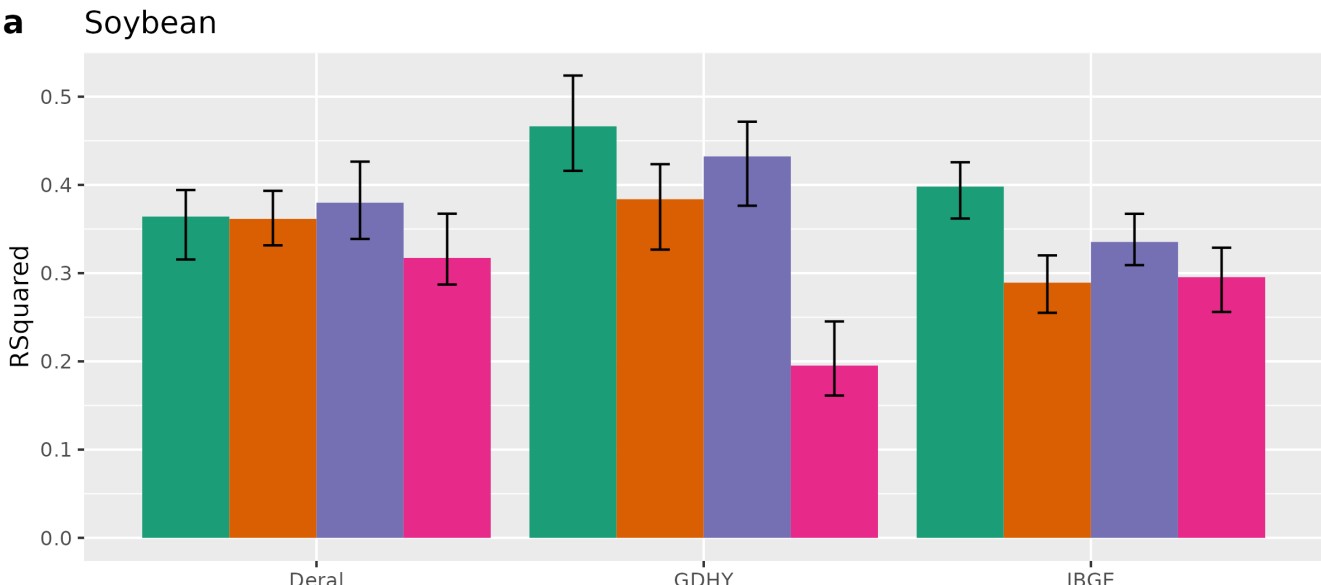

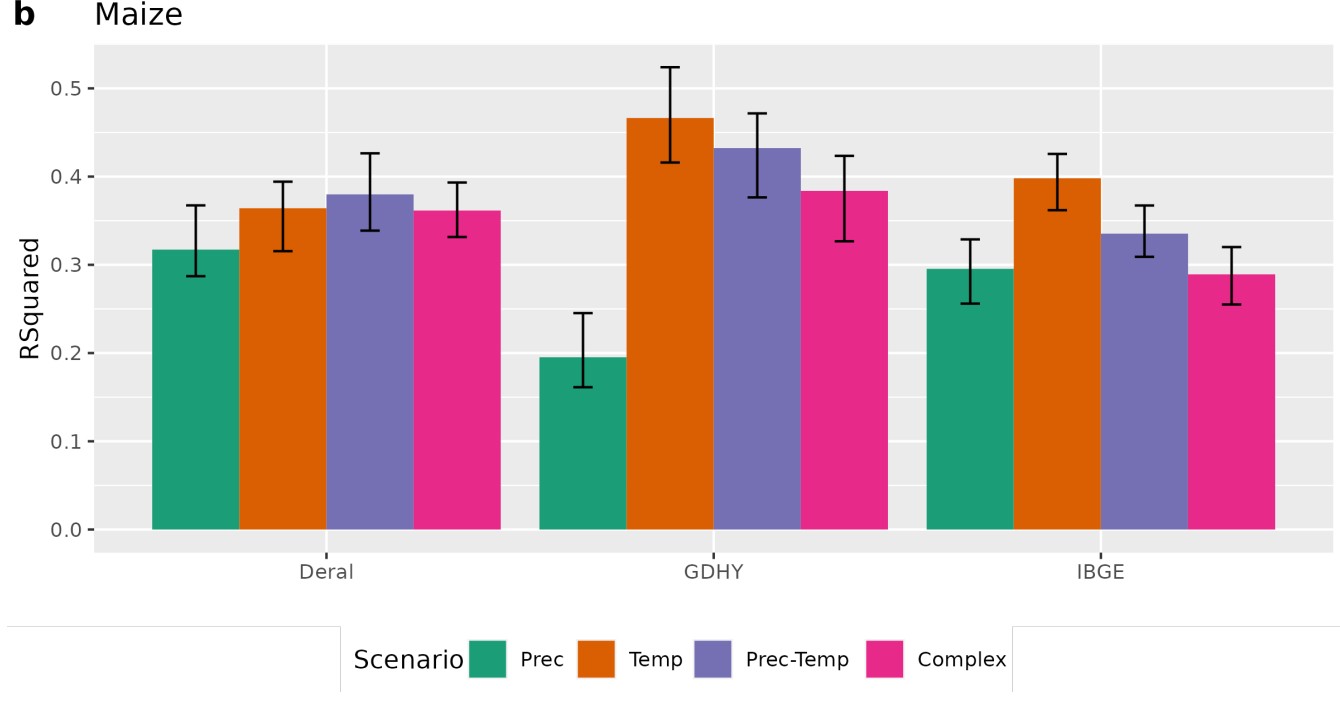

**Figure 4.** Performance evaluation of regression model for (a) Soybean and (b) Maize. Considering data from Department of Rural Economics (Deral) with data only for Paraná, Global Dataset on Historical Yields (GDHY) and Brazilian Institute of Geography and Statistics (IBGE) for the major historical agricultural producing municipalities





The model performance was generally higher for GDHY than for the other datasets for soybeans in the southern states (RS, SC, and PR). The models based on precipitation means and the combination with temperature and extremes explain the variability of crop yield more than only the temperature means, which was observed in all three datasets. For maize second season, for MS, MG, and GO, the models that combine mean temperature to mean precipitation and extremes explained more the variability of crop yields than only temperature, which was observed in the GDHY and IBGE datasets. The IBGE database
showed that the maize model was much more effective in São Paulo than in any other state.

The random forest models used in this study helped obtain the most relevant variables, and these variables were classified into CID types, i.e., wet and dry, hot and cold, and subcategories, as described in Table 2. The model demonstrates the importance of climate variables in explaining the variability in crop yields and allowed us to determine the critical climatic impact-drivers considering each state and dataset. The following sections summarize the key CID for soybeans and maize.
Figures demonstrating the most important variables for soybean in Fig. 5 and for maize in Fig. 6.





**Figure 5.** Variable Importance Analysis in Random Forest for Soybean Yield Prediction Across Brazilian States. The figure illustrates the variable importance scores obtained from Random Forest models applied to three distinct datasets: Deral, IBGE, and GDHY, encompassing seven Brazilian states - RS, SC, PR, SP, MS, MG, and GO







**Figure 6.** Variable Importance Analysis in Random Forest Models for Maize Yield Prediction Across Brazilian States. The figure illustrates the variable importance scores obtained from Random Forest models applied to three distinct datasets: Deral, IBGE, and GDHY, encompassing seven Brazilian states - PR, SP, MS, MG, and GO





### 3.1.1 Wet and dry

Changes in mean precipitation pose a threat to agricultural production. Precipitation deficit leads to reduced available soil moisture, affecting plant development and reducing crop yields, and it is considered the most critical environmental factor that reduces crop yields (Bray, 2007). In our analysis, mean precipitation was one of the most important climatic impact-driver on

soybean crop yields during January and February for RS, SC, PR, MS, and also in December in PR.

Interpretations The state of Rio Grande do Sul has historically been affected by El Niño Southern Oscillation (ENSO), with a more substantial influence from November to May (Gelcer et al., 2013), which is responsible for droughts and impact on soybean crop yields. The model did not indicate that mean precipitation was the most important for the state of SP. For maize production, mean precipitations during April and May were considered important for PR, MS, MG, and GO, except for SP.

Agricultural systems require minimum rainfall, or they rely on irrigation. In Brazil, SP, MS, MG, and GO states have a well-defined difference between wet and dry seasons. Usually, the wet season starts in October and ends in May, and the Soybean-Maize double cropping system depends on the length of the wet season in the states mentioned above.

Agricultural and ecological drought indices are directly related to a precipitation deficit and excessive temperature (Lesk et al., 2022; Sarhadi et al., 2018; Lesk et al., 2021), which affects the ability of plants to grow and reduce plant transpiration.

The duration and timing of droughts play a significant role. We observed that droughts occurring in January and February were the most important for soybeans. The droughts in February and March also affected the second-season crop yields of maize. Droughts that occurred at the end of the maize growing season also affected crop yields.

In this study, we considered climate extreme indices on different time scales. As we added the temporal dimension to the analysis, we revealed that a 3-month SPEI in October in the state of RS was selected on the list of most relevant variables.

This indicates that pre-sowing meteorological factors that can reduce soil moisture conditions also influence crop yields. This result corroborates Santini et al. (2022), which revealed that drought analysis should not neglect antecedent conditions since it influences factors such as soil workability and crop development.

### 3.1.2 Hot and cold

The mean air temperature influences many aspects of crop cultivation. In RS and SC, the soybean growing season starts

when the mean temperatures exceed the minimum temperature thresholds for soybeans (Battisti and Sentelhas, 2014). As the temperature increases, the development (phenology) of the plant is affected, and increased thermal stress is expected Lesk et al. (2021). Except for RS, mean temperatures during all soybean growing seasons were considered important variables. The same behavior was observed for maize; mean temperatures were considered significant during all growing seasons.

Exposure to temperatures above a specific limit or threshold can lead to lower yields. The value of these thresholds depends

on the crop species and farm management. For soybeans, extreme temperature indices affected crop yields throughout the growing season, especially in January and February.





## 3.2 Determining Thresholds and Their Significance

With the results of the selection of critical climatic impact-divers, we improved the understanding of the impacts on climate variables that significantly influence crop yield losses, considering different types of indices and critical periods. The in-
sights obtained from the combination of random forest models applied to different datasets facilitate a robust understanding of climate-crop interactions and make it possible to compare what results the datasets have in common, increasing the results' reliability. However, the random forest model did not provide information on the values of each variable that are important and can help us define the threshold values of these indicators that are associated with an increased risk of crop yield losses.

To improve our understanding of how each climate extreme indicator was used to predict, we used SHAP. This technique
allowed us to extract insights by coupling the results of a Random Forest model with those of an XGBoost model, thus providing a comprehensive perspective on the drivers of crop yield fluctuations. The results of SHAP-derived explanations revealed a clear pattern concerning the most influential variable that affects soybean yields. We highlight critical loss events by evaluating the model prediction for a particular city in a given year.

In 2019, an important widespread drought event was observed in Brazil and was considered a mega-drought that affected
many regions of Brazil, especially the Paraná river basin Marengo et al. (2021). We highlight the model explanation for the state of Paraná considering two important agro-producing municipalities, namely Borrazópolis and Uniflor, as shown in Fig. 7.

The two datasets presented an agreement regarding crop yields below the expected value $E[X]$. However, they varied in terms of the magnitude of these losses. For the municipality of Borrazópolis, the Deral yields were lower than IBGE, 1.53 and 3.27, respectively. The predictions made by the two models were similar, and the main variables also performed similarly.
High temperatures, represented by the maximum value of the daily minimum temperature in February, were the main driver of losses combined with the 3-month SPEI in February.

For Deral, accumulated precipitation in December was also a significant driver of losses, and for IBGE, 3-month SPEI in January. The other variables represented a negligible influence on crop yield losses. For the city of Uniflor, the actual yields of Deral and IBGE were similar, 2.59 and 2.85, respectively. The main influences on crop yield losses were precipitation in
December (prctot_Dec) and high temperatures (tnx_Feb). The 3-month SPEI values in January and February can be considered redundant. Standardized indices refer to previous conditions; therefore, the values overlap in two months (December and January).



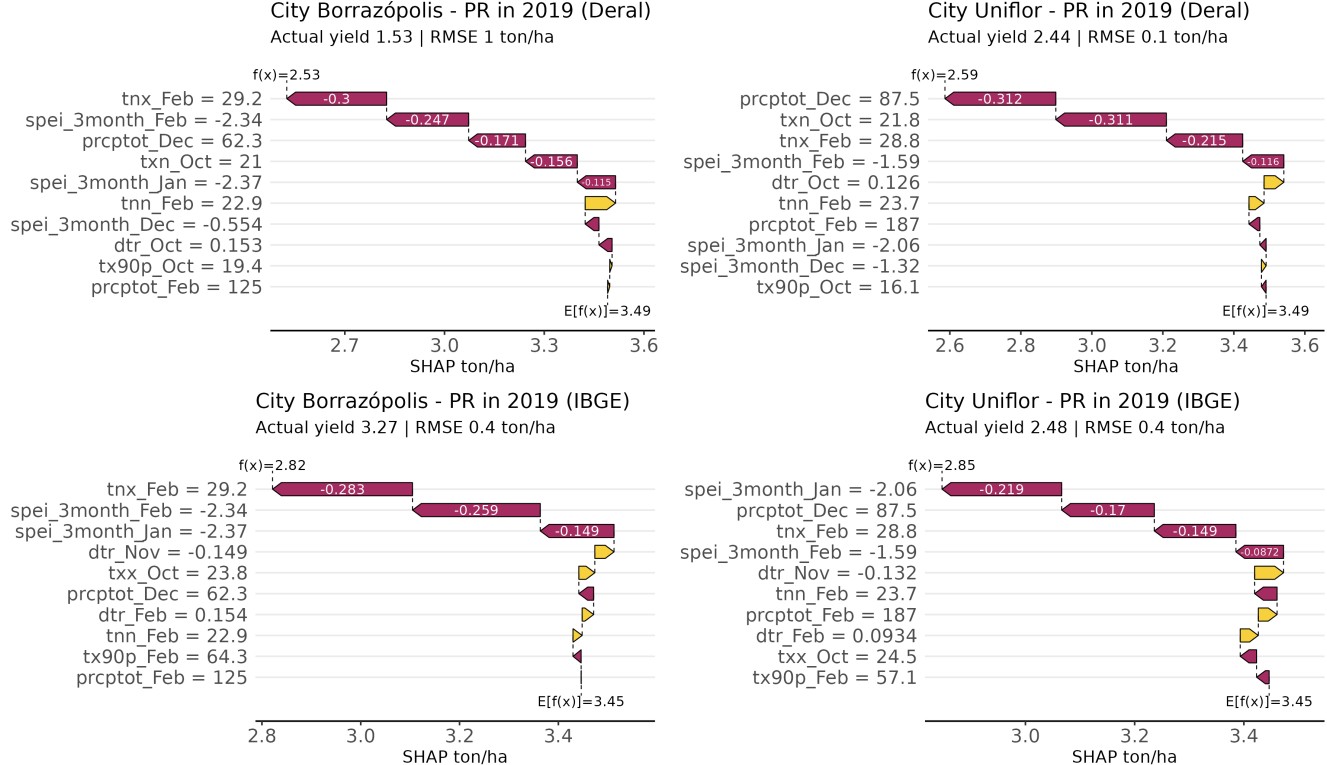

**Figure 7.** SHAP waterfall plot visualizing the key climatic impact-drivers contributions to crop yield losses for the state of Paraná (PR) in 2019, a drought year.Monthly Maximum Value of Daily Minimum Temperature (tnx), Minimum Value of Daily Maximum Temperature (txn), Minimum Value of Daily Minimum (tnn), Maximum Value of Daily Maximum (txx), Percentage of days when TX > 90th percentile, Standardized Precipitation Evapotranspiration Index (SPEI), Total Precipitation (prctot), Daily temperature range (DTR)

In 2014/2015, a severe drought occurred in southeastern Brazil, causing an unprecedented water supply shortage in the Cantareira Water Supply System and affecting many cities in São Paulo (Deusdará-Leal et al., 2019). The drought had reper-

cussions in many regions of the state of São Paulo. Therefore, we compared the results of IBGE and GDHY for two munici-palities of the state of So Paulo, Orlândia and Pontal. The expected values of GDHY were lower than those of IBGE, and the two datasets diverged in yields below the expected value, i.e., IBGE indicated losses, and GDHY did not. According to a report by the Brazilian Ministry of Agriculture, Livestock and Food Supply (MAPA), the state of São Paulo was one of the most affected by losses in the agricultural year 2014/2015 (MAPA, 2022). This result suggests that, although it has been suggested

that GDHY is recommended in data-scarce regions (Iizumi and Sakai, 2020), using this dataset requires caution.



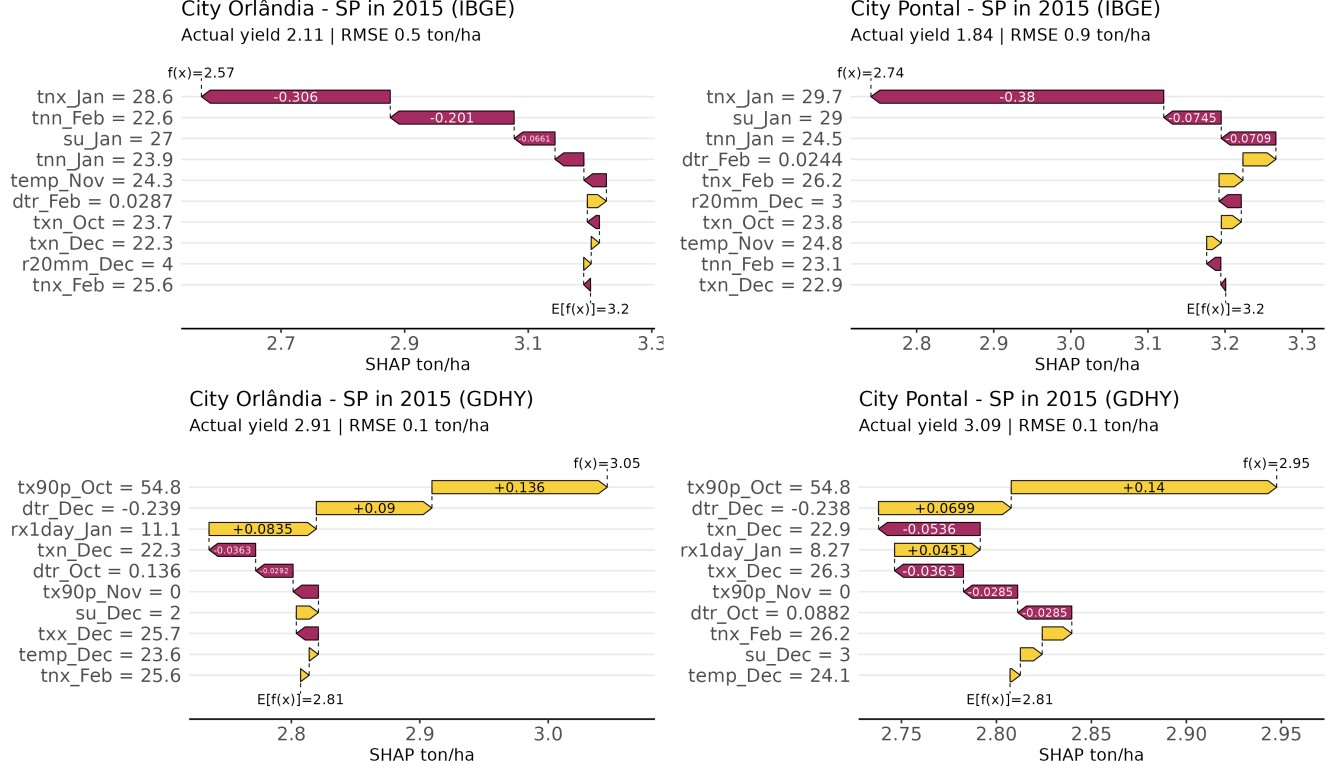

**Figure 8.** SHAP waterfall plot visualizing the key climatic impact-drivers contributions to crop yield losses for the state of São Paulo (SP) in 2015, a drought year. Monthly Maximum Value of Daily Minimum Temperature (tnx), Minimum Value of Daily Maximum Temperature (txn), Minimum Value of Daily Minimum (tnn), Maximum Value of Daily Maximum (txx), Percentage of days when TX > 90th percentile, Standardized Precipitation Evapotranspiration Index (SPEI), Total Precipitation (prctot), Daily temperature range (DTR)

The SHAP methodology analyzes each prediction and shows how each variable was used in the model. This helps us create a partial dependence plot, which relates the variable's value with the impact in terms of crop yield losses represented by the SHAP value. This analysis is illustrated in Fig. 9, which shows partial dependence plots for the state of Rio Grande do Sul.

We compared two datasets. The IBGE data set shows that the 3-month SPEI in February can influence crop yield losses
and has an upper and lower threshold. The lower threshold is -1.5. Values below this can represent losses of up to 0.8 tons/ha. Generally, 3-month SPEI values below -1.5 are considered critical and are used as a reference for the severity of the drought (Chiang et al., 2021). The upper limit indicates that extreme wet conditions also affect crop yields in the RS state. We find threshold 1.

Excess rainfall can have the same impact as droughts (Li et al., 2019). However, little attention has been paid to this analysis.
Our results suggest excessive precipitation can be responsible for up to 0.4 ton/ha of losses. More studies are suggested on this type of hazard. The cumulative precipitation in February presented a threshold of around 100 mm and the potential to cause losses of approximately 1 ton/ha. The cumulative precipitation in December was the only indicator with a similar result in the



IBGE and GDHY datasets. Regarding potential losses, both agree on a value of up to 0.8 ton/ha; however, the for IBGE is 150 mm, and for GDHY, it is 100 mm.





**Figure 9.** A comparison of the key climatic impact-drivers derived from the Brazilian Institute of Geography and Statistics (IBGE) from 2013 to 2021 and the Global Dataset on Historical Yields (GDHY) from 2009 to 2016 annual data aggregated at the municipal level was used to create a dependence plot for the soybean explanation model for validation data in RS. The data spanned from 2013 to 2021 for IBGE and from 2008 to 2016 for GDHY.. Considering 3-month SPEI in February (spei_3month_Feb) for (a) IBGE and (b) GDHY; precipitation accumulated in February (prcptot_Feb) for (c) IBGE and (d) GDHY, and precipitation accumulated in December (prcptot_Dec) for (e) IBGE and (f) GDHY



The patterns of crop yield losses observed in the region raise two main concerns. The first is that the severe crop yield losses presented in the previous examples have happened only once in the entire time series, representing an imbalance in the values of the data set. One implication of this situation is that models may not have sufficient cases of severe failure to be trained adequately and may underestimate losses. The second concern is related to the decision to do with these anomalous events. Possible solutions include using it for training, testing, or removing it from the dataset. We opted to keep these events in the
analysis with the warning that this might interfere with the model performance. However, we wanted to evaluate the ability of the model to predict unprecedented loss events.

### 3.3  Evaluating Combined Hazards

The SHAP algorithm also allowed us to investigate the compound effect of climate indicators. In Fig. 10, we exemplify the detection of compound event effects, considering the most important variable in the state of PR (prcptot_Dec) with four
other considered important variables. We observed hot and dry compound events, a combination of high temperatures and a precipitation deficit. Compound hot and dry events have been cited as an increasing threat to food production (Lesk et al., 2022; Zscheischler et al., 2018; Hamed et al., 2021).

This analysis showed that precipitation in December was closely related to a 3-month SPEI in January. As discussed previously, this is expected since SPEI considers previous conditions. However, it is essential to note that December is a critical
month for droughts in PR and other states, such as Rio Grande do Sul (RS), Santa Catarina (SC), and Mato Grosso do Sul (MS) (see Section 3.3.2.1).

Interestingly, indices based on the minimum daily temperature best reflected the impact of hot days. The maximum value of the daily minimum temperature in February (tnx_Feb) presented critical values of 27.7 °C, and the minimum value of the daily minimum temperature in February (tnn_Feb) presented a critical value of 22 °C. When minimum temperatures are high,
it is likely that maximum temperatures are also high, and the difference between minimum and maximum daily temperatures is small; this is a possible explanation of why the daily temperature range (dtr_Oct) has a negative impact on crop yields when its values are close to zero.

Our use of XGBoost with SHAP explanation provides an advance by enabling quantification of the combined effect of multi-hazards on food production. In the realm of risk for food production, this method could be applied to explain the seasonal impact
forecast made with composite indicators such as the Integrated Drought Index (IIS) (Cunha et al., 2018; Marengo et al., 2017). This method can also be readily applied to other natural hazards, such as landslides, floods, and wildfires.

### 4  Conclusions

This study aimed to assess the impacts of climate extremes on food production using explainable machine learning algorithms. To achieve this goal, we extensively examined various datasets, focusing on soybean and second-season maize in Brazil. Our
data sources included the Department of Rural Economy, the Brazilian Institute of Geography and Statistics (IBGE), and the Global Dataset for Historical Yields (GDHY). Through a machine learning analysis, we examined the effects of climate





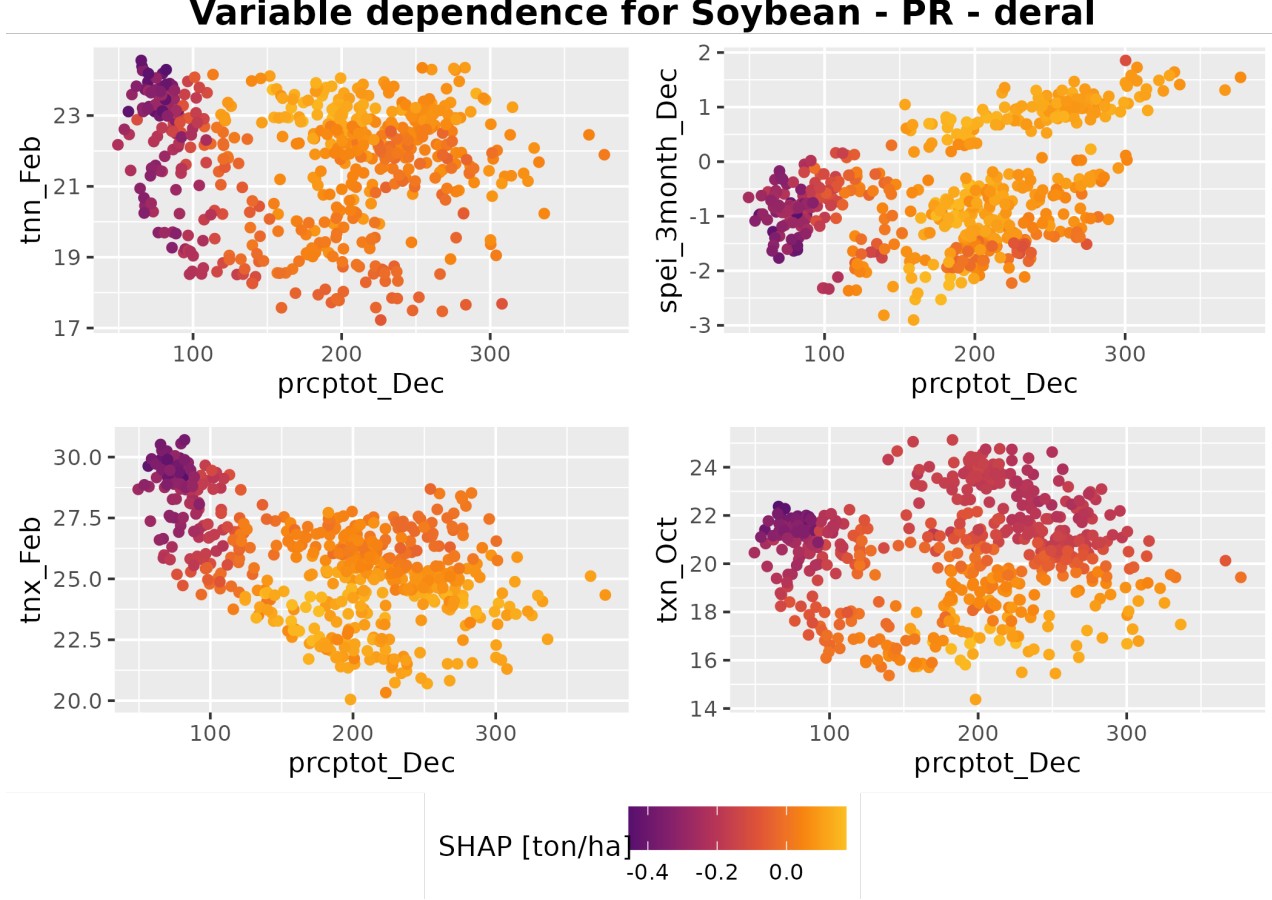

**Figure 10.** 2D partial dependence derived from from the Department of Rural Economics (Deral) of the state of Parana from 2016 to 2021 of the key climatic impact-driver total precipitation in December (prctot_Dec) with the most correlated CID and their combined impact on SHAP values (ton/ha): daily temperature range in October (dtr_Oct); Considering 3-month SPEI in Jan (spei_3month_Jan); Daily Minimum Temperature in February (tnx_Feb), Minimum Value of Daily Minimum (tnn_Feb)

extremes on crop yield production, ultimately providing critical insights for the agricultural sector. Our analysis incorporated data from several Brazilian states, including RS, SC, PR, SP, MS, MG, and GO for soybeans and PR, SP, MS, MG, and GO for the second season of maize.

We conducted two machine models to achieve our research objectives. In the first model, we explored different combinations of input data, encompassing precipitation and temperature means, and more complex combinations, including precipitation, temperature means, and extremes. This approach allowed us to determine the most relevant climate indices for the investigated regions. In particular, this experiment validated the robustness of our methodology, as it successfully identified climate indices of particular significance for regional studies.

We took the most relevant indices from the first experiment in the second model and used them in an XGBoost model. We then applied Shapley Additive Explanations (SHAP) explanatory analysis to explore how the random forest model utilized the important indices to predict the impact of climate extremes on food production. This analysis revealed the impact of these indices and provided insights that may be crucial in establishing significant thresholds and guidelines for effective climate-driven decision-making.

In conclusion, our research exemplifies the potential of machine learning to understand and harness the influence of climate variables on food production. By determining the most pertinent CIDs and exploring their significance in a regional context, our findings contribute to a growing body of knowledge critical for informed decision-making, policy development, and adaptive strategies in the face of climate change and its impact on agriculture. As demonstrated in our study, the combination of data-driven insights and advanced modeling techniques offers a valuable pathway toward ensuring food security under climate

change.

*Author contributions.*

Conception and design of the work: MRB, RFS, GCG. Data collection, manuscript drafting: MRB; Discussion and analysis: MRB, RFS, GCG, PAAM, and EMM. Critical review of the manuscript: MRB, RFS, GCG, AMS, ALCBD, PAAM, and EMM. Advisor: EMM.

*Competing interests.*

The contact author has declared that none of the authors has any competing interests.

*Acknowledgements.* We wish to express our appreciation to the reviewers and editors for their valuable feedback and contributions to this project. We would like to thank the University of São Paulo for providing a stimulating research environment and resources that made this work possible. The authors also Acknowledge that the generative AI technology ChatGPT 3.5 was used only and solely to minor text

corrections for correcting grammar and improving readability.



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
