# Peer review of "A data-driven framework for assessing climatic impact-drivers in the context of food security"

_EGUsphere, 2023_

## Author Comment (AC1)

Author comments (AC) **(in bold and blue)** to the referee comments (RC) **(*in bold and italics*)**
* * *
*Shortly about myself to better interpret my review: I am an agricultural economist postdoc fellow, working in the interdisciplinary field of agricultural trade, food security, with application of econometrics and machine learning, mostly interpretable. I have no in-depth background in climate change.*

**AC: We thank referee 2 for reviewing our manuscript. His/her field of expertise is important for our work, because we used historic climate data to predict impact on crop yield variability using machine learning algorithms. Although Climate Change is a critical challenge, we have not addressed it directly in our manuscript. As a response to this comment, we will make sure to thoroughly review the manuscript to make sure that the aspect of climate change is better explained.**

*Summary:*

*The paper uses the Climatic Impact-Driver (CID) approach to evaluate the impact of climate risks on food security, focusing on maize and soybeans in Brazil. The authors use data-driven methods and machine learning models to identify the most relevant climate indices and their thresholds that increase the impact probability. They found that mean precipitation is a key CID, with specific thresholds indicating increased risk of crop yield losses. The study emphasizes the relevance of both extreme and mean climate indices in assessing climate risk on agriculture, contributing to decision-making and policy development in response to climate change.*

**AC: We thank referee 2 for summarizing our work and providing the key aspects of what we've done. While we are happy that we were able to convey a clear message, we would benefit from learning more about his/her technical judgment about our manuscript.**

*Introduction*

*I find the introduction comprehensive and insightful, providing a clear overview of the challenges associated with predicting crop yield variability in response to climate extremes. The emphasis on the importance of considering multiple weather variables and employing models that incorporate sector-specific vulnerability and exposure adds depth to the discussion, highlighting the complexity of agricultural risk assessment. The introduction's exploration of machine learning algorithms, particularly decision tree algorithms like random forest models, offers innovative possibilities for improving predictive accuracy despite data availability constraints.*

*Furthermore, I like the idea of using model interpretability techniques in the modeling framework to address the limitations of existing approaches. The integration of the CID framework promises a solid foundation for contextualizing climate in decision-making, aligning with the need for localized solutions in agricultural systems. Overall, the introduction effectively sets the stage for a research endeavor that holds significant potential for informing critical decisions and strategies aimed at enhancing food production resilience in the face of climate variability.*

**AC: We thank you again for the comment from referee 2. We truly value the feedback and will make sure to take the suggestions into account during our review process.**

*Methodology*

*This methodology section presents a comprehensive approach to investigate the impacts of climate extremes on soybean and maize crop yields in Brazil, which is of paramount importance for agricultural research and policy-making. The modeling framework outlined, with its emphasis on data filtering, variable selection, and threshold determination, offers a systematic way to analyze the complex relationships between climatic variables and crop yields. By integrating different interpretable machine learning techniques, the study ensures both predictive accuracy and interpretability, crucial for gaining trust of people who will later use the proposed modelling-framework.*

*The delineation of the study area and selection criteria for municipalities provide a clear understanding of the geographical scope and rationale behind the dataset selection. Moreover, the detailed description of data collection and processing, including the handling of outliers and missing values, enhances the reliability and reproducibility of the study's findings. Additionally, the inclusion of soil data enriches the analysis by considering the influence of soil properties on crop productivity.*

*However, while the methodology appears robust and well-structured, some sections could benefit from further clarification. As a non-expert in climate change, I could benefit from an explanation regarding the application of climate indices and their relevance to crop yield analysis. Providing more insights into the selection process of specific indices and their interpretation within the context of agricultural impacts would enhance the understanding of the wide audience. Overall, this methodology sets a solid foundation for investigating climate extremes' impacts on food production, contributing valuable insights to the field of agricultural economics.*

**AC: We thank referee 2 for the feedback. We selected the framework of climate impact-drivers (CID) from IPCC to increase the standardization of our work. Nonetheless, we have not evaluated explicitly the impacts of climate change on crop yields using (CIDs). The main idea of the work is to use explainable machine learning algorithms to select the most relevant CIDs for the crops selected in different geographic regions of Brazil. This process can be used to help decision makers to select climate indicators for impact modeling and monitoring. The applications can be**

**extended for crop yield losses forecasts, parametric insurance design and risk analysis. I am not really sure how referee 2**

*Results and Discussion*

*The chapter is clear and summarizes the article very well.*

AC: We thank referee 2 again for taking time to review our manuscript. The comments are mostly summarizing the main idea of the paper and pointing out the aspect of climate-change in the methodology. We understand that climate change plays a fundamental role in risk analysis for crop yields, however, evaluating long term projections was not the main goal of our paper. We focused on using a data-driven method with historic climate data and crop yield data to select climate impact-drivers (CID).

---

## Author Comment (AC2)

Author comments (AC) **(in bold and blue)** to the referee comments (RC) (***in bold and italics***)
* * *
*General comments:*

*The study uses interpretable machine learning to identify the most important climate impact drivers for predicting maize and soybean yield variability in Brazilian states. Overall, the manuscript is quite well-written and has clear descriptions of the datasets used which are very helpful for the reader. They discuss in some detail the advantages and disadvantages of the use of different yield datasets in the region, which is crucial for the interpretation of the results of this type of analysis, and make the effort to show a comparison of the datasets and where they agree and disagree. They also use two specific examples of droughts in Brazil as case studies to examine the interpretations, which is interesting and helps to verify their approach.*

*The topic is very important, and novel methods such as this have a clear use-case in identifying the most relevant periods at which different CIDs impact yields. However, I have some concerns about the methodology. The description of the methodology used is not sufficiently thorough, so these concerns may have been addressed by the authors, but this should be clarified.*

**AC: We are deeply thankful for the comments of reviewer 1. We will make sure to carefully address each of the suggestions to clarify the concerns about the methodology, so it can be accepted for publication. The suggestion to use spatiotemporal correlation in the cross-validation strategy was the major contribution from reviewer 1 for our work and we will be glad to implement and document this change in our methodology. We are confident that the changes will make our work stronger and increase the impact of the publication in the research community. We provide a more detailed response for each comment in the paragraphs below.**

*Random forests are often used for this type of study and are a good choice when working with tabular data such as this. However, care must be taken when using any machine learning method not to allow the model to overfit to dependencies or correlation between features. The training and testing method used was not explained clearly, except in Figure 1, which only states that 20% of the data was used to test the models, but not how that 20% was selected. Given that models were trained on a state level, multiple municipalities within each state would have highly correlated climate and yields. Were the data points split in time and/or space to account for this, or sampled randomly? If they were sampled randomly, this can lead to misleading estimations of model performance and the interpretations are less likely to represent the physical mechanisms that are intended to be studied. Particularly relevant - if soil is used as a predictive feature, which does not vary in time in the dataset used (I believe), the model can easily spatially overfit.*

*Overall, I find the manuscript to be quite well-written and the thorough analysis of the different datasets used and how they impact the results is interesting and excellent scientific practice. However, I think that some small changes to the methodology (most importantly, selecting a test set considering the spatiotemporal autocorrelation and estimating SHAP values using this test set,*

*ideally using a different feature selection method such as SFS) and better explanation of the steps involved to generate the results discussed could very much improve the paper. As the paper aims to present a framework to enhance the interpretability of ML methods fo crop yield loss prediction, it is important that the framework is robust and can deal with common issues for this type of problem such as overfitting to spatiotemporal data.*

AC: We thank referee 1 very much for this comment. Overfitting is a major concern when working with random forests. Each municipality has, at best, ca. 40 data points. This means that there is not enough data for a data-driven model at municipal level. The choice of pooling data at state level is a way of using a priori knowledge to group data. We are pretty aware that this strategy should be further evaluated. However, we also acknowledge that pooling data at state level is an acceptable strategy, especially for policy making. Nearby municipalities can be highly correlated, therefore we understand the concern of referee 1 and the need to further clarify this in the methodology. The papers and methods suggested by referee 1 were very helpful for us to elucidate this issue in our methods.

*Finally, given that the title of the paper and stated goal is to present a framework that can be used by other researchers, the code used should be published and made openly available, but this is not currently stated in the manuscript.*

AC: We definitely agree with this comment. We are organizing all the scripts to be shared in a repository (e.g., GitHub). This will be appended to the manuscript for the next round of review.

*Specific comments:*

*At what stage was RFE used to select features, and how was this conducted? How many features were selected? I also question the use of RFE in cases where models can overfit (e.g. when spatiotemporal data is used), as features that the model find most important are more likely to not be physically meaningful. Using, for example, sequential feature selection with a spatial or temporal cross-validation splitting method would be more likely to return relevant drivers, and I would recommend to the authors to try this if possible.*

AC: Thanks for the suggestions, despite RFE being frequently used for reducing the number of variables, we have not applied this method in our study. We are sorry for that and will make sure to clarify this in the text. In this paper, instead of using RFE, we applied a random forest model to find the 10 most important variables and then we used these variables as input for the SHAP model. After careful consideration of the spatiotemporal overfitting model, we decided to perform important methodological adaptation to clarify this matter and improve the reliability

of our results. These changes can be easily implemented in the R Codes we've already have and should NOT be time costly.

1. Creating spatial blocks considering spatial autocorrelation of crop yields using Valavi et al. (2019) that applies Roberts et al. (2017) cross-validation strategy. From this, we can create spatial blocks that incorporate autocorrelation of crop yields, temperatures and precipitation means over municipalities. Here, we present an example that considers that autocorrelation has a range of 87 km for the Deral dataset in the Brazilian state of Paraná.

[Figure]

[Figure]

2. Spatio-temporal k-fold Cross-validation using the method of nearest neighbor distance matching (Mila et al 2022, Linnenbrink et al 2023) available for implementation in R using the CAST package (Meyer et al., 2024).
3. Training a random forest model using the caret package (Kuhn, 2022);
4. Removing variables that cause overfitting using forward feature selection. It is important to say that this step will substitute the need of selecting only the 10 most important variables;
5. Using the most important variables in the SHAP framework.

*In Figure 4, it would be helpful to have descriptions of what features were included in the different scenarios - in particular, I could not understand what 'Complex' meant.*

AC: We thank referee 1 for the comment and we will make sure to add the descriptors suggested in Figure 4. There are multiple ways to derive multi-hazard scenarios for crop yields. As a way to promote a simplification focusing on improving the explainability of the different models, we decided to sub-divide input climate drivers according to the hazard type, i.e., precipitation means only, temperature means only, precipitation and temperature means, and

**precipitation means, temperature means and extreme indices. The combination of mean weather variables and extreme weather variables was called 'Complex'. We will make sure to make it more clear in the methodology section describing the scenarios using a table.**

*In Figure 5 and 6, is this after RFE has been used to select only 10 features? I was confused by the fact that for maize, only February features are shown, but later in the text it states that April and May precipitation was important for some regions.*

**Thank you for your comment. In fact, the maize second cycle growing season starts in February, therefore, the beginning of the growing season plays an important role in determining the crop yields. We should make it clear in the discussion of the results.**

*I would strongly advice not removing correlated variables before doing the feature selection. You can expect that the highly correlated variables will not both be selected, and it is another opportunity for data leakage to enter.*

**We appreciate the comment and agree that we can avoid removing correlated variables, since we perform random feature elimination, we should use all the variables in our study avoiding data leakage.**

*I think it is very useful to compare the importances between the different states and datasets, as this can help to find robust insights and identify potential problems with the datasets used. It would be useful to see uncertainty quantification here as well, as given that similar model performance can come from many combinations of features (as shown in Figure 4), one would expect that there is significant uncertainty in the interpretations as well. I would also consider using an additional feature importance metric (permutation feature importance on held-out test set?) for comparison, but this might be out of scope.*

**We agree with the comment and think that the held-out test set could help us to improve the discussion with uncertainty quantification. While this suggestion can provide additional feature importance metrics, we believe that this could be explored in future work.**

*I also find it unusual to fit random forest models and then to use a more complex model (XGBoost) to explain them via SHAP. Normally, SHAP is used directly on the trained model to be interpreted, and if a second model was used it would normally be a simpler model. Why not use XGBoost for the initial part of the analysis instead of adding this complexity of using a second model to explain the first?*

**Thank you for your comment. We agree that this choice should be better explained in the manustrip. SHAP uses game theory to analyze the impact of variables considering the interaction with other variables. In other words, Shapley values estimate the importance of a**

feature by contrasting the model prediction without that feature. The computation cost of adding all the variables makes running the model in a modern microcomputer impractical. Moreover, since we are decomposing the indices in monthly steps, in order to make the *post hoc* analysis feasible, we need to reduce the dimensionality of input variables. We understand that choosing 10 variables as a prior definition can exclude important variables. Since we will perform a permutation feature importance, we can use a posteriori criteria such as the input variables that represent ca. 80% of model variability.

[Figure]

*Partial dependence plots do not need SHAP values - they can be calculated by just varying individual features and estimating the output. It might be interesting to compare this against those gained from SHAP (but again, maybe out of scope). It would at least be useful to discuss/justify in the text why the partial dependence plots gained from SHAP are more useful (which is very plausible).*

We thank the reviewer for the comment. Partial dependence plots were used in other papers (e.g., Vogel et al., 2019) that were cited in our manuscript. We agree that it is an interesting idea to compare the two plots to understand what insights can be gained by the use of SHAP partial dependence plots, however, we believe that this comparison should be included in future work.

*SHAP values are also sensitive to the data used to calculate them, and I would again recommend to use test sets for this that are split with consideration to the spatial and temporal correlations.*

We thank you for the comment. As described previously, the spatial and temporal correlations will be taken into consideration for the improved model and we believe that this will significantly improve the reliability of our results.

*Interpreting the results of this type of study can be difficult, as in general, any feature used for training is one that could be a causal driver. This means that it is hard to figure out if the results are meaningful or if the model has learned some spurious correlations. The fact that only February features are shown as important for maize suggests, to me, that something strange is going on, as the authors state that this is peak planting date and in some regions, planting is not finished until the beginning of April. It seems more likely that heat, for example, would be more important during the reproductive period. Using the different test sets as I mentioned before might help with this, as well as using instead of the internal RF variable importance measure.*

**We agree with the reviewer that interpreting the results is difficult, especially considering the complex systems like climate and agriculture. We need to review these results. To enhance the robustness and interpretability of our results, we will apply the permutation feature importance test in addition to the current methodology. This adjustment will help validate our findings and provide a clearer understanding of the causal relationships driving maize yields, particularly in the context of varying climate conditions.**

*Why remove heteroskedasticity? Could this be justified more in the text? As we expect more climate variability with climate change and therefore more yield variability, it isn't obvious that this should be corrected for.*

**We thank the reviewer for this comment and we believe that the influence of climate change on crop yield variability is a complex topic. We based the decision to remove heteroskedasticity on the actuarial literature (Tolhurst and Ker, 2014; Liu and Ker, 2020; Osaki et al., 2008). The aforementioned papers demonstrate the importance of removing heteroskedasticity and that it can be also related to technological changes.**

*Lines 171-172 describe a second analysis using Gaussian copulas, but I could not find this further described or any results from this in the rest of the manuscript?*

**In the manuscript, we mention a second analysis that employs Gaussian copulas to evaluate the combined effect of variables. This analysis complements the analysis using SHAP values to interpret the contributions of individual features. SHAP values provide a way to quantify the contribution of each feature to the prediction made by a model. The combination of SHAP values and Gaussian copulas, allowed us to further evaluate the dependencies between features. In other words, we evaluated the effect of one feature might be influenced by the presence or value of another feature. We agree with the reviewer that this segment should be better explained both in the methodology section and in the results section.**

*Technical corrections:*

*The paragraph on interpretability (lines 53 to 56) I could not understand.*

**We thank the reviewer for the comment. We agree that the paragraph needs to be improved both in terms of logic and language.**

**Old version:**

*The paradigm of interpretability of machine learning models is a broad topic of discussion in supervised learning (Lipton, 2018). Two essential observations related to model interpretability are: (i) and (ii) the training data can be imperfect to represent a dynamic environment that changes over time.*

**New version:**

*The paradigm of interpretability of machine learning models is a broad topic of discussion in supervised learning (Lipton, 2018). The model interpretability can be achieved by means of feature engineering and using interpretable models such as linear models, that is "algorithmic transparency". When the features, or input data, are decomposed and the number of variables make the interpretation of models difficult, post hoc interpretation can be used to extract explanations from learned models.*

*Please state briefly that the crop yields were detrended in the main text (the further explanation in the Supplementary is very helpful, but there is no mention of the fact that the yields are detrended in the main manuscript which is very important to interpret the results).*

**We agree with the comment. We can bring the main information that was handed in the supplementary material to the main text to better explain**

*Some references on selecting test sets appropriately when using ML with spatiotemporal data:*

*Meyer, H., Reudenbach, C., Wöllauer, S. & Nauss, T. Importance of spatial predictor variable selection in machine learning applications – Moving from data reproduction to spatial prediction. Ecological Modelling 411, 108815 (2019).*

*Sweet, L., Müller, C., Anand, M. & Zscheischler, J. Cross-Validation Strategy Impacts the Performance and Interpretation of Machine Learning Models. Artificial Intelligence for the Earth Systems 2, (2023).*

*Roberts, D. R. et al. Cross-validation strategies for data with temporal, spatial, hierarchical, or phylogenetic structure. Ecography 40, 913–929 (2017).*

We thank for the references suggested by the reviewer and we believe that adopting the spatial and temporal aspects mentioned in the text will considerably improve the impact of the manuscript.

References

Kuhn. caret: Classification and Regression Training. R package version 6.0-93. 2022. Available at https://CRAN.R-project.org/package=caret.

Meyer et al.. CAST: 'caret' Applications for Spatial-Temporal Models. R package version 1.0.1 2024. Available at https://CRAN.R-project.org/package=CAST.

Valavi R, Elith J, Lahoz-Monfort JJ, Guillera-Arroita G. blockCV: An R package for generating spatially or environmentally separated folds for k-fold cross-validation of species distribution models. Methods Ecol Evol. 2019; 10:225–232. doi: 10.1111/2041-210X.13107.

---

## Author Response (AR1)

Replies (**in bold and blue**) to the comments/suggestions (*in italics*)

**Dear Editor Aloïs Tilloy,**

**Thank you very much for considering our manuscript "A data-driven framework for assessing climatic impact-drivers in the context of food security" for publication in Natural Hazards and Earth System Sciences. We appreciate the time and effort you and the reviewers have invested in evaluating our work.**

**As requested, we have thoroughly revised our manuscript to address the reviewers' suggestions. We have ensured that all necessary clarifications have been made to enhance the manuscript's clarity and have referenced similar analyses to provide a more comprehensive context for our work.**

**We also revisited and evaluated the model codes to guarantee that the problem of spatio-temporal overfitting is clarified. All the scripts used in the manuscript are now available at the GitHub repository https://github.com/marcosrbenso/ClimateImpactML. We will explain what were the main changes in the point by point-by-point response to the reviewers comments.**

**Moreover, we invited prof. José Marengo, who is a well-known expert in the field of climatology to help us to validate our results.**

**We believe these revisions have significantly strengthened our manuscript, making our research more transparent and its contributions clearer. Please see below for our detailed responses to all the referees' comments and suggestions. We look forward to your feedback on the revised manuscript.**

**Best regards,**

**Marcos Roberto Benso**

**On behalf of the authors**

**Reviewer #1**

*General comments:*

*The study uses interpretable machine learning to identify the most important climate impact drivers for predicting maize and soybean yield variability in Brazilian states. Overall, the manuscript is quite well-written and has clear descriptions of the datasets used which are very helpful for the reader. They discuss in some detail the advantages and disadvantages of the use of different yield datasets in the region, which is crucial for the interpretation of the results of this type of analysis, and make the effort to show a comparison of the datasets and where they agree and disagree. They also use two specific examples of droughts in Brazil as case studies to examine the interpretations, which is interesting and helps to verify their approach.*

*The topic is very important, and novel methods such as this have a clear use-case in identifying the most relevant periods at which different CIDs impact yields. However, I*

*have some concerns about the methodology. The description of the methodology used is not sufficiently thorough, so these concerns may have been addressed by the authors, but this should be clarified.*

*Random forests are often used for this type of study and are a good choice when working with tabular data such as this. However, care must be taken when using any machine learning method not to allow the model to overfit to dependencies or correlation between features. The training and testing method used was not explained clearly, except in Figure 1, which only states that 20% of the data was used to test the models, but not how that 20% was selected. Given that models were trained on a state level, multiple municipalities within each state would have highly correlated climate and yields. Were the datapoints split in time and/or space to account for this, or sampled randomly? If they were sampled randomly, this can lead to misleading estimations of model performance and the interpretations are less likely to represent the physical mechanisms that are intended to be studied. Particularly relevant - if soil is used as a predictive feature, which does not vary in time in the dataset used (I believe), the model can easily spatially overfit.*

*Overall, I find the manuscript to be quite well-written and the thorough analysis of the different datasets used and how they impact the results is interesting and excellent scientific practice. However, I think that some small changes to the methodology (most importantly, selecting a test set considering the spatiotemporal autocorrelation and estimating SHAP values using this test set, ideally using a different feature selection method such as SFS) and better explanation of the steps involved to generate the results discussed could very much improve the paper. As the paper aims to present a framework to enhance the interpretability of ML methods fo crop yield loss prediction, it is important that the framework is robust and can deal with common issues for this type of problem such as overfitting to spatiotemporal data.*

**Thank you for your detailed and constructive feedback on our manuscript. We have addressed your concerns in our revised manuscript as follows:**

1. **To avoid problems with temporal dependencies with data, we split the data into the first 80% datapoints for training and the last 20% datapoints for testing. For the training datasets, we applied a leave-one-year-out cross-validation approach (LOYOCV) (L99-L105)**
2. **The soil dataset was disconsidered for the model since it can easily spatially overfit.**
3. **We need to clarify that the 10 most important features were selected using feature importance. That is, we applied the feature importance rank that resulted from a random forest model and selected the 10 most important features.**

*Finally, given that the title of the paper and stated goal is to present a framework that can be used by other researchers, the code used should be published and made openly available, but this is not currently stated in the manuscript.*

Replies **(in bold and blue)** to the comments/suggestions (*in italics*)
* * *
**After revising the codes to make sure to address the feedback given by Reviewer #1, all the scripts used in the manuscript are now available at the GitHub repository https://github.com/marcosrbenso/ClimateImpactML.**

Specific comments:

*At what stage was RFE used to select features, and how was this conducted? How many features were selected? I also question the use of RFE in cases where models can overfit (e.g. when spatiotemporal data is used), as features that the model find most important are more likely to not be physically meaningful. Using, for example, sequential feature selection with a spatial or temporal cross-validation splitting method would be more likely to return relevant drivers, and I would recommend to the authors to try this if possible.*

**We thank for the suggestions, we believe that using sequential feature selection with a spatial or temporal cross-validation could be very interesting for future work. For this manuscript, we used, as mentioned previously, LOYOCV for selecting the best fit model and extract feature importance and selected the 10 most important features. We believe that this helped to clarify the concerns about the methodology. (L99-L105)**

*In Figure 4, it would be helpful to have descriptions of what features were included in the different scenarios - in particular, I could not understand what 'Complex' meant.*

**After careful consideration, we decided to remove this figure from the text. We opted for showing the model performance of the best-fit model.**

*In Figure 5 and 6, is this after RFE has been used to select only 10 features? I was confused by the fact that for maize, only February features are shown, but later in the text it states that April and May precipitation was important for some regions.*

**We improved the representation of the best features selection showing a new figure that represents the most important feature according to the month. This helps to improve the discussion of the physical meaning of our method of variable selection. Moreover, the complete list of most important features was added in the supplementary material.**

*I would strongly advice not removing correlated variables before doing the feature selection. You can expect that the highly correlated variables will not both be selected, and it is another opportunity for data leakage to enter.*

**Thank you for your comment. Despite the fact that we acknowledge that data leakage is an important issue, features that have a correlation higher than 0.9 considering pearson correlation coefficients tend to be too similar, therefore, redundant. We still think that is important to remove.**

*I think it is very useful to compare the importances between the different states and datasets, as this can help to find robust insights and identify potential problems with the datasets used. It would be useful to see uncertainty quantification here as well, as given*

*that similar model performance can come from many combinations of features (as shown in Figure 4), one would expect that there is significant uncertainty in the interpretations as well. I would also consider using an additional feature importance metric (permutation feature importance on held-out test set?) for comparison, but this might be out of scope.*

*I also find it unusual to fit random forest models and then to use a more complex model (XGBoost) to explain them via SHAP. Normally, SHAP is used directly on the trained model to be interpreted, and if a second model was used it would normally be a simpler model. Why not use XGBoost for the initial part of the analysis instead of adding this complexity of using a second model to explain the first?*

**To answer the two aforementioned comments, we opted to use only a random forest model for the two modeling steps, that is, selecting the 10 most important features and then explaining the impact of the 10 most important features on crop yield prediction. For the SHAP model, we also divided into training and testing. The choice of RF models is due to the fact that this model has been widely used in the literature for many environmental applications, specially for crop yield studies. Also, XGBoost calibration can be tricky with a large number of features. That's why we used the random forest model for selecting the most important features and then applied the XGBoost. Nonetheless, to avoid confusion, in this new version of the manuscript we used only RF models.**

*Partial dependence plots do not need SHAP values - they can be calculated by just varying individual features and estimating the output. It might be interesting to compare this against those gained from SHAP (but again, maybe out of scope). It would at least be useful to discuss/justify in the text why the partial dependence plots gained from SHAP are more useful (which is very plausible).*

**We thank the reviewer for the comment. Partial dependence plots were used in other papers (e.g., Vogel et al., 2019) that were cited in our manuscript. We agree that it is an interesting idea to compare the two plots to understand what insights can be gained by the use of SHAP partial dependence plots, however, we believe that this comparison should be included in future work.**

*SHAP values are also sensitive to the data used to calculate them, and I would again recommend to use test sets for this that are split with consideration to the spatial and temporal correlations.*

**As aforementioned, we split the data into testing and training for SHAP model as well.**

*Interpreting the results of this type of study can be difficult, as in general, any feature used for training is one that could be a causal driver. This means that it is hard to figure out if the results are meaningful or if the model has learned some spurious correlations. The fact that only February features are shown as important for maize suggests, to me, that something strange is going on, as the authors state that this is peak planting date and in some regions, planting is not finished until the beginning of April. It seems more likely that heat, for example, would be more important during the reproductive period.*

Replies (**in bold and blue**) to the comments/suggestions (*in italics*)
* * *
*Using the different test sets as I mentioned before might help with this, as well as using permutation feature importance instead of the internal RF variable importance measure.*

**We agree with the reviewer that interpreting the results is difficult, especially considering the complex systems like climate and agriculture. We tried to improve the discussion and clarity of the feature selection both in the manuscript and the supplementary material by displaying the full table of variable importance tests.**

*Why remove heteroskedasticity? Could this be justified more in the text? As we expect more climate variability with climate change and therefore more yield variability, it isn't obvious that this should be corrected for.*

**After careful consideration, we decided to remove this figure from the text. We opted for showing the model performance of the best-fit model. (L87-L88)**

*Lines 171-172 describe a second analysis using Gaussian copulas, but I could not find this further described or any results from this in the rest of the manuscript?*

**We agree with the reviewer that his part of the paper was poorly explained. Therefore, we improve the description of this method by including more details of what library was used and how this analysis is performed (L125-L135)**

*Technical corrections:*

*The paragraph on interpretability (lines 53 to 56) I could not understand.*

**We thank the reviewer for the comment. We agree that the paragraph needs to be improved both in terms of logic and language.**

**Old version: The paradigm of interpretability of machine learning models is a broad topic of discussion in supervised learning (Lipton, 2018). Two essential observations related to model interpretability are: (i) and (ii) the training data can be imperfect to represent a dynamic environment that changes over time.**

**New version: The paradigm of interpretability of machine learning models is a broad topic of discussion in supervised learning (Lipton, 2018). The model interpretability can be achieved by means of feature engineering and using interpretable models such as linear models, that is "algorithmic transparency". When the features, or input data, are decomposed and the number of variables make the interpretation of models difficult, post hoc interpretation can be used to extract explanations from learned models.**

*Please state briefly that the crop yields were detrended in the main text (the further explanation in the Supplementary is very helpful, but there is no mention of the fact that the yields are detrended in the main manuscript which is very important to interpret the results).*

**Thank you for your comment. The detrend of time series is now mentioned in the L185-L187**

Replies **(in bold and blue)** to the comments/suggestions (*in italics*)
* * *
**Reviewer #2**

**We thank Reviewer #2 for taking the time to read and post feedback for our manuscript. There is no substantial suggestion in the message given by Reviewer #2, therefore, we believe that, with the changes made according to Reviewer #1, we encompassed all changes necessary to improve the manuscript. However, we are looking forward to hearing further comments from reviewer #2.**

**Reviewer #2**

---

## Author Response (AR2)

**Dear Editor Aloïs Tilloy,**

**Thank you very much for considering our manuscript for publication and for doing diligent work as editor. We are happy with how Reviewer #1 is engaged in improving our manuscript and for the throughout revision that was performed.**

**We have revised our manuscript to answer Reviewer's #1 comments. We ensured to answer all the confusion points and feedback regarding the methodology. We believe that with the revised manuscript, the reader will be much more confident in reproducing the method we proposed. We also made sure to correct all technical comments to improve readability and avoid typos and small writing mistakes.**

**Please see below for our detailed responses to all the referees' comments and suggestions. We look forward to your feedback on the revised manuscript.**

**Best regards,**

**Marcos Roberto Benso**

**On behalf of the authors**

**Reviewer #1**

First, are models trained individually for each spatial point? This is not explicitly stated anywhere, as far as I can tell.

**Thank you for raising this point. To clarify, we create one model for each particular state. Specifically, for each state, we trained separate models using each of the three datasets (IBGE, Deral, and GDHY). This means that, for each state, we have three distinct models corresponding to the three datasets.**

**To address this more explicitly, we have added the following clarification to Section 2.1:**

***"Different models are independently trained for each state being analyzed, a separate and unique machine learning model is developed and trained using data specific to that state. This implies that the analysis of climatic impact-drivers (CID) on food production is customized to account for the unique characteristics, data, and conditions present in each state, rather than applying a single model uniformly across all states."***

It is also not clear from the text how the CV and training/test evaluation is executed. For example, lines 151-152 (in the file with tracked changes): 'The creation of training, validation, and testing subsets is crucial to avoid overfitting and achieve reasonable estimates of model performance. The data set was divided into the first 80% for training and 20% for validation data.'. This does not explicitly state that the split is in time, and although a training, validation and test set is mentioned, in fact only a training and test split is done.

Thank you for your observation. We acknowledge the need for greater clarity regarding the splitting strategy. To clarify:

The training and validation datasets were split considering the chronological order of the data. Specifically, the first 80% of the time was allocated for training, and the last 20% was reserved for validation. This temporal split ensures that the model is tested on later, unseen data, in a way that happens in the real-world predictive scenarios. While we referred to the second subset as "validation data" in the manuscript, this term might have contributed to the misunderstanding, as no separate test dataset was used in this workflow.

To address this, we have revised the relevant text (lines 151–152) as follows:
*"The first 80% of the data, according to the timeline, was used to train the model, allowing it to learn and adjust its parameters. The remaining 20% was used for validation, meaning this portion was reserved to test the model's predictions on data it hasn't seen during training. This approach, which incorporates a temporal aspect, is intended to simulate a real-world scenario where future data should be predicted. This method helps prevent overfitting by ensuring that the performance of the model is evaluated on new unseen data that come after the training period used, thus providing a realistic assessment of how the model will perform in practice."*

Additionally, it is unclear how the LOYO-CV is done - the text states a 'fixed window' of 10 years was used, and one year as a test set; is this repeated such that every year is a validation set and then the resulting scores are averaged, as is usually done for CV? 'Fixed window' reads as if only a single split is done, but I assume the authors do not mean that.

Thank you for raising this point. We recognize that our description of the LOYO-CV methodology was not sufficiently detailed, which may have led to ambiguity. To clarify:

In the LOYO-CV approach, we employed a fixed window of 10 years for training, followed by a single year as the test set. This process was repeated iteratively, leaving each year out as the test set while using the preceding 10 years for training. The evaluation metrics were computed for each iteration, and the final scores were averaged across all iterations to provide a robust estimate of model performance.

To avoid misunderstanding, we have revised the text to provide greater clarity as follows:
*"A fixed 10-year window was used for training, followed by one year as a test set. This process was repeated iteratively, leaving each year as the test set while using the preceding 10 years for training. Performance metrics were calculated for each iteration and scores were averaged to obtain an overall assessment of the performance of the model."*

Line 160: 'the best fit model was determined by employing a leave-one-year-out cross-validation approach (LOYOCV)': I do not understand this step. Did the authors train multiple models, with identical hyperparameters and features, on different subsets of years and select the best one according to its performance on each corresponding validation set? This would not be advisable, as the validation set performance would depend on the year, not just the model performance. Or were the hyperparameters selected or features selected based on the average CV performance across all folds, as would be more typical?

**The hyperparameters were selected based on the average CV performance across folds. Thank you for observing this point. To improve the clarity of this point, we added the following text in the manuscript:**

***"the best-fit model was selected using a leave-one-year-out cross-validation method (LOYOCV), and hyperparameters were chosen according to the mean CV performance in folds"***

Lines 175-177: 'Different models were trained considering different combinations of input data, including precipitation means, temperature means, and combinations of means and extreme climate indices. The goal of this experiment was to identify the most important climate indices.' I would recommend describing this more explicitly. Which subsets of features were tested and how were they evaluated?

**I think this paragraph does not represent what was done in the manuscript we presented. To avoid confusion, we decided to remove it from the text.**

I would suggest that the authors explicitly state which years are covered in the training and test sets, on which years the LOYO-CV is done, and further on in the text, on which years/data the results are calculated on for each figure. This should also be included in Figure 1.

**Thank you for your feedback. We believe the explanation regarding the training/validation split and the cross-validation (CV) procedure was clear. However, we understand that the issue arises from the datasets differing in length and the years they encompass, which may have caused some confusion.**

Additionally, there are multiple ways of calculating feature importance from random forest models. I am assuming that the authors refer to the internal entropy-based feature importance. This should be stated explicitly. I would also advise that the authors consider additionally calculating the permutation-based feature importance on the validation or test years. Their agreement or disagreement with the entropy-based feature importance would aid readers in assessing the robustness of the findings.

**Thank you for your comment. We believe that comparing the agreement and disagreement considering permutation-based feature importance is a great idea. Nevertheless, after careful evaluation, we decided to clarify the method we used as Reviewer #1 suggested and perhaps compare the entropy-based with permutation-based feature importance for future work.**

*"The feature importance was determined based on entropy is determined by calculating the reduction in entropy (information gain) each feature provides when used to split the data at each node in the decision tree. Features that result in greater reductions in entropy across the tree are considered more important."*

**Other points of confusion:**

- Line 128, 129 introduces 'explainable' and 'operational' features, but does not define them. Also lines 137-8: 'Other relevant aspects, such as relevancy, explainability, and operationability, will be explained in the following steps.' As far as I can see, these terms aren't explained later in the text.

**The idea behind mentioning the terms 'explainable' and 'operational' features represent the motivation of the methodology. We believe that, by using such a method that we proposed, we can achieve that. To avoid misunderstanding we clarified this in the text:**

*"In this study, the concepts of "explainable" and "operational" features are the motivation for our proposed methodology. We aim to achieve a balance between model performance, interpretability, and practical applications. By focusing on explainable features, our objective is to create models that offer clear insights into decision-making processes, thereby promoting transparency and reliability. This interpretability is essential for stakeholders who must comprehend and validate the model's results."*

**For simplicity and to avoid confusion, we decided to remove the passage " Other relevant aspects, such as relevancy, explainability, and operationability, will be explained in the following steps." from the text.**

- Line 144: 'The SHAP method uses a second model, most commonly the RF model...' I do not think this is true. SHAP is used to explain the original model.

**Thank you for pointing it out. We corrected it in the text by reformulating the sentence:**

*"The SHAP method is used to explain how each variable was used to make each prediction"*

- Line 162: 'The models were trained and optimized on the training and validation datasets' This is unusual, did the authors intend to write only training datasets, not validation datasets?

**Thank you for noticing, this was only applied to the training dataset. We corrected it in the text**

- Outlier removal: Lines 276-277 state that the authors remove outliers using the interquartile range, but from the supplementary material it seems that only few datapoints are removed, which does not make sense. Additionally, the supplementary material discussion of the outlier removal is confusing: in lines 30-31 they state 'Removal of outliers is a complex problem since we are working with extreme events', but this is followed by an explanation of the trend and heteroskedasticity removal process, and not the outlier removal. Then, in line 52: 'After obtaining a consistent time series corrected for outliers, trends, and heteroskedasticity' - but the outlier removal occurs afterwards, as far as I can tell (line 59: 'To eliminate potential outliers, we excluded values considering each year and state').

**I think that this issue requires clarification. We considered the immediate regions that consist of a group of municipalities that are close together. The definition of immediate region is given by IBGE. The crop yields in these regions were very similar based on the correlation index of the yields. Then, we considered the interquantile range of each year. For example, in a give year, if the yield of one municipality within the immediate region is much higher or lower than the other, this would be considered an outlier.**

**In the main text we added the following explanation:**

**For the outlier removal process, we defined "immediate regions" as clusters of municipalities geographically proximate to one another, as classified by the IBGE. Crop yields within these regions exhibited a high degree of correlation, which was verified using the correlation index. To identify outliers, we applied the interquartile range (IQR) method for each year. Specifically, if the yield of a municipality in a given year deviated significantly from the yields of other municipalities in the same immediate region, it was classified as an outlier and excluded from the dataset. This approach ensured that only extreme and anomalous data points, not reflective of regional trends, were removed.**

**In the Supplementary material**

*"After obtaining a consistent time series corrected for trends, heteroskedasticity, and outliers"*

*"To eliminate potential outliers, we excluded values considering each year and immediate region within the state. This was done because he hypothesizes that within the immediate region, the crop yields should be similar"*

- Lines 277-278: 'Changes in technology in seed production, fertilizers, and land management, also known as technological trends (Liu and Ker, 2020) were removed by Local Polynomial Regression Fitting (LOESS)' - all trends would be removed, including those due to e.g. climate change, not just technology. I would recommend mentioning this.

**Thank you for pointing this out. We added it in the text:**

*"and other sources of trend such as climate change were removed by Local Polynomial Regression Fitting (LOESS)"*

- Supplementary material line 137: the sentence ending in 'indicating the significant role of both rainfall' is incomplete.

**We completed the sentence:**

**_"indicating the significant role of both rainfall and temperature."_**

- Figure 3: Where do the error bars come from here? Which variables were used as predictive features? Are these metrics calculated on the test set?

**Since different models were trained, we used the performance of the models trained for each state to create the error bars.**

- Lines 340-342: Where do these ranges come from?

**These ranges come from Figure 5. To clarify, we added the reference in the text.**

- Lines 349-354: Where are the results discussed here shown?

**The results are the analysis of Figure 7. We added the reference and improve the writing of this passage of the manuscript.**

- Figure 4: The hazard types don't correspond, as far as I can tell, to the CID types and categories from Table 2. What do these labels mean?

**Thank you for pointing this out. We actually meant CID categories. We corrected the figure.**

- Line 420: 'This technique allowed us to extract insights by coupling the results of a Random Forest model' - I don't think SHAP works by coupling a model to another, it is intended to explain the original model.

**We agree that this sentence is a bit confusing. We corrected in the text:**

**_"This technique allowed us to extract insights from the results of a Random Forest model"_**

- In lines 466-468, and in the Supplementary material (lines 95-101) the authors discuss whether or not to keep the most extreme years in the training or test set, or remove them. I understand from the text that they were kept in the dataset, but it doesn't say if they were used in the test or training set.

**All the extreme years were kept in the dataset, we mentioned it in the supplementary material.**

**_"As our aim was to assess the effects of extreme climate events, we opted to retain all of these extreme events within the dataset"_**

**Technical corrections:**

- Line 136: I would remove the sentence 'Feature selection is a pre-processing step in machine learning models'. It is confusing as the feature selection is described later in the text.

**We agree on this correction and we removed it from the manuscript.**

- 'ML' is introduced as an abbreviation early in the text, but the authors continue to use 'machine learning' afterwards. I would also advise using 'RF' as an abbreviation for random forest to improve readability.

**In fact, this correction will improve the readability. We kept only the first time the terms "random forest" and "machine learned" appeared. After that, we only used RF and ML as suggested.**

- Figure 1 is helpful, but there are multiple typos and minor formatting issues. E.g. 'Filter Highly correlated variables' should be 'Filter highly correlated variables'; 'Boostrap RF model' should be 'Bootstrap RF model'.

**Thank you for the correction. We applied it to the figure 1**

- Line 170: 'To achieve this, we used the Random Forest model' This is repeated multiple times in the text, and could be removed.

**Thank you for the correction. We removed it from the text.**

- Line 175: 'Different models were trained considering different combinations of input data' - I believe the authors refer to different combinations of features or variables, rather than data.

**The reviewer #1 is correct. We referred to different combinations of features. However, we also need to add the we considered different crop yield datasets.**

- Line 207: Typo - 'The  SHAP explanations was performed'

**The typo was corrected**

- Lines 239, 240, 244: Maize and soybean should not be capitalised here.

**This was corrected.**

- Figure S1 and S2: Typo - eath should be each
**This was corrected**

- Supplementary line 86: The reference is erroneously capitalised: RODRIGUES et al. (2013)

**This was corrected**

- Supp line 130: typo - Table SS2 should be S2

**This was corrected**

- In the Supplementary material, Section 4 still refers to an XGBoost model.

**This was corrected**

- Lines 364-365: Typo - 'The analysis can be of variable importance for soybean datasets is shown in Table S1'

**This was corrected**

- Line 413: Typo - 'climate impact-divers' should be 'drivers'

**This was corrected**

- Lines 461-462: Typo - 'however, the for IBGE is 150mm'
**This was corrected**

- Figure 7: Please include the units for e.g. precipitation.

**This unit for precipitation was included in the figure**

- Line 471: 'exemplify' is the wrong word, I think - perhaps 'present'?

**Reviewer #1 is correct. The word "present" is more suited. Corrected it in the text.**